# Multiscale modelling of desquamation in the interfollicular epidermis

**Claire Miller**[1,2,3¤], **Edmund Crampin**[2,4,5†], **James M. Osborne**[1]*

**1** School of Mathematics and Statistics, The University of Melbourne, Parkville, Australia, **2** Systems Biology Laboratory, School of Mathematics and Statistics, and Department of Biomedical Engineering, The University of Melbourne, Parkville, Australia, **3** Computational Science Lab, Informatics Institute, University of Amsterdam, Amsterdam, Netherlands, **4** School of Medicine, Faculty of Medicine, Dentistry and Health Sciences, The University of Melbourne, Parkville, Australia, **5** ARC Centre of Excellence in Convergent Bio-Nano Science and Technology, Melbourne School of Engineering, The University of Melbourne, Parkville, Australia

† Deceased.
¤ Current address: Auckland Bioengineering Institute, The University of Auckland, Auckland, New Zealand
* jmosborne@unimelb.edu.au

**Data Availability Statement:** The source code used to produce the data and analyses presented in this manuscript are available on GitHub: https://github.com/clairemiller/MultiscaleModellingDesquamationInIFE.

## Abstract

Maintenance of epidermal thickness is critical to the barrier function of the skin. Decreased tissue thickness, specifically in the stratum corneum (the outermost layer of the tissue), causes discomfort and inflammation, and is related to several severe diseases of the tissue. In order to maintain both stratum corneum thickness and overall tissue thickness it is necessary for the system to balance cell proliferation and cell loss. Cell proliferation in the epidermis occurs in the basal layer and causes constant upwards movement in the tissue. Cell loss occurs when dead cells at the top of the tissue are lost to the environment through a process called desquamation. Desquamation is thought to occur through a gradual reduction in adhesion between cells, due to the cleaving of adhesion proteins by enzymes, in the stratum corneum.

In this paper we will investigate combining a (mass action) subcellular model of desquamation with a three dimensional (cell centre based) multicellular model of the interfollicular epidermis to better understand maintenance of epidermal thickness. Specifically, our aim is to determine if a hypothesised biological model for the degradation of cell-cell adhesion, from the literature, is sufficient to maintain a steady state tissue thickness. These investigations show the model is able to provide a consistent rate of cell loss in the multicellular model. This loss balances proliferation, and hence maintains a homeostatic tissue thickness. Moreover, we find that multiple proliferative cell populations in the basal layer can be represented by a single proliferative cell population, simplifying investigations with this model.

The model is used to investigate a disorder (Netherton Syndrome) which disrupts desquamation. The model shows how biochemical changes can cause disruptions to the tissue, resulting in a reduced tissue thickness and consequently diminishing the protective role of the tissue. A hypothetical treatment result is also investigated: we compare the cases of a partially effective homogeneous treatment (where all cells partially recover) and a totally effective heterogeneous treatment (in which a proportion of the cells totally recover) with the

**Funding:** This research was supported by an Australian Government Research Training Program (RTP) Scholarship (awarded to CM), and in part conducted and funded by the Australian Research Council Centre of Excellence in Convergent Bio-Nano Science and Technology (project number CE140100036, awarded to EC). The funders had no role in study design, data collection and analysis, decision to publish, or preparation of the manuscript.

**Competing interests:** The authors have declared that no competing interests exist.

aim to determine the difference in the response of the tissue to these different scenarios. Results show an increased benefit to corneum thickness from the heterogeneous treatment over the homogeneous treatment.

## Author summary

We are interested in understanding how the thickness of the epidermis—the outer layer of the skin—is maintained. Maintaining a sufficient thickness is critical to skin, and whole body, health. To do this, we combine mathematical models of processes occurring to and between cells in the epidermis, and the processes occurring at a smaller scale that drive them. This model can help us to understand the relationship between the two scales and their role in regulating epidermal thickness. More specifically, it is broadly accepted that the loss of skin cells from the top of the epidermis is due to a gradual reduction in the adhesion between connecting cells as they move upwards through the epidermis. We look in further detail at what is causing this decrease in adhesion, and at the balance between the loss of adhesion, resulting cell loss, and creation of new cells in the lower layers of the epidermis. This balance is what determines epidermal thickness. We also consider how the disease Netherton Syndrome affects the adhesion degradation, and implement hypothetical treatment outcomes to better understand the system.

## Introduction

The thickness of the epidermis varies both between body location and individual, with averages in the range of 50 to 100 $\mu$m measured in human studies [1, 2]. Sufficient epidermal thickness is critical to skin, and whole body, health. The epidermis is in a state of constant cell turnover, fully renewing every 4 weeks [3]. Consequently, homeostatic tissue thickness is maintained through a balance of cell input, via proliferation, and output, through a process called desquamation. The behaviour we focus on in this paper is desquamation, with the aim to improve our understanding of how it is controlled by subcellular processes in order to maintain tissue thickness.

Cell proliferation occurs in the basal (bottom-most) layer of the epidermis. There are several hypotheses as to the proliferative cell lineage existing in this layer. Early cell lineage models hypothesised a stem cell population dividing to produce one stem and one transit amplifying cell, which further divided 3 times before differentiating to a non-proliferative cell [4]. More recent models propose proliferative cell populations which divide symmetrically or asymmetrically with a certain probability. This has been proposed for both a single progenitor population [5] and a stem and progenitor population that behave in a similar way but at different rates (4–6 stem cell divisions per year compared to 1 progenitor division per week) [6]. Alternatively, Sada et al. [7] hypothesise 2 progenitor populations, one dividing symmetrically and one asymmetrically at different rates (1.4 and 3.5 divisions per week respectively).

The other layer critical to cell turnover is the most superficial layer of the epidermis: the stratum corneum. In the stratum corneum, cells are very flat (<0.5 $\mu$m) and wide (20–40 $\mu$m), compared to the almost spherical shapes seen in the basal layer [8–10]. It is from the top of the stratum corneum that cells are lost through a process known as desquamation. Desquamation refers to the shedding of the cells in clumps (squames) from the surface of the skin, in part due to a reduction in cell-cell adhesion towards the top of the tissue. Excessive desquamation of the

epidermis causes deficiencies in barrier function of the skin. This allows the ingress of anti-gens, resulting in inflammation, and has been linked to allergic diseases such as atopic dermatitis (eczema), food allergies, or asthma [11, 12]. Diseases of the skin can range from a nuisance to a major health risk, with debilitating or even fatal consequences, and even the milder diseases can have follow on effects for our health. One such disease, which we use as an example in this paper, is Netherton Syndrome (NS). This disease affects the subcellular desquamation process, causing premature desquamation, skin peeling, frequent skin infection, and temperature instability [13].

## The subcellular biology of desquamation

A critical part of the desquamation process is the degradation of cell-cell adhesion molecules. In the lower layers of the epidermis, cell-cell adhesion is due to adhesion protein complexes called desmosomes. When the cell transitions into the stratum corneum, the desmosomes are converted to a more specialised adhesion protein complex: corneodesmosomes, through the addition of corneodesmosin. As the cell proceeds upwards through the stratum corneum, these corneodesmosomes are degraded via an assortment of proteases (enzymes that break down proteins) [14, 15]. The transition and degradation of cell-cell adhesion is shown in Fig 1.

As illustrated in Fig 1, the degradation of the corneodesmosomes is not uniform around the circumference of the cell: it occurs first in the vertical direction on the horizontal surfaces of the flattened corneum cells, and then in the planar direction. This delayed degradation in the

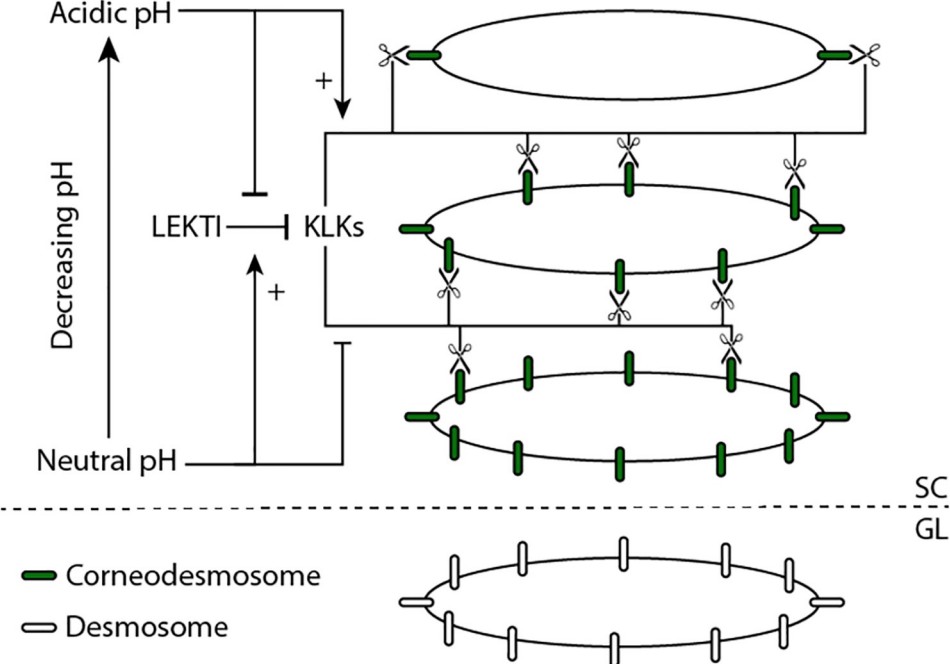

**Fig 1. A diagram representing the involvement of pH, LEKTI, and KLKs in the desquamation process.**
Desmosomes (white ellipses) are converted to corneodesmosomes (green ellipses) as they enter the stratum corneum (SC in figure) from the granular layer (GL in figure). These corneodesmosomes are then degraded by KLKs, first on the horizontal surfaces then the vertical (indicated by the scissors). The degradation process is inhibited by LEKTI, which binds with the KLKs and stops the KLKs from binding with the corneodesmosomes. The local pH regulates the inhibition, and the corneodesmosome degradation rate by KLKs. Arrows with '+' symbols indicate the pH increases the reaction, while '⊣' indicates the pH reduces the rate of the reaction.

planar direction is potentially due to the presence of other adhesion protein complexes called tight junctions limiting the access of the proteases to the corneodesmosomes [16].

The specific protease subgroup that is known to degrade the proteins in the corneodesmosomes is kallikrein serine proteases, or KLKs [11, 12, 15, 17]. These KLK enzymes are released by the cell into extracellular space at the base of the stratum corneum (via lamellar bodies) [12, 14]. Other families of proteases are also known to contribute to degradation, however we only consider KLK proteases as they are thought to be the primary enzyme, and are consequently the most studied [12, 18, 18, 19]. The data available on this process is mostly limited to KLK5 and KLK7 [12, 18]. KLK5 constitutes around 10% of the proteases and KLK7 constitutes 36%. However, KLK5 is a trypsin-like KLK while KLK7 is chymotrypsin-like. The remaining 54% of KLKs are all trypsin-like, and therefore we assume KLK5 has properties more representative of the majority than KLK7. As a result, for simplicity, we focus on KLK5 in this study and assume it is representative of all KLK protease activity.

The final components of the system shown in Fig 1 are the inhibitor LEKTI and the pH gradient. In addition to KLK proteases, the lamellar bodies also secrete LEKTI which is an inhibitor to the reaction between the KLK proteases and the corneodesmosomes [13, 19, 20]. The effectiveness of LEKTI at inhibiting the KLK-corneodesmosome reaction is regulated by the local pH, which varies vertically through the stratum corneum. The pH gradient changes from neutral in deep corneum (6.8 in women, 6.9 in men) to acidic in superficial corneum (5.3 in women, 4.5 in men) [21]. At neutral pH (deep corneum) there is a high level of interaction between LEKTI and KLK, while low pH (superficial corneum) increases dissociation of LEKTI and KLK, allowing for higher rates of corneodesmosome degradation [12].

In addition to regulating the LEKTI-KLK interaction, the pH also directly contributes to regulation of the KLK-corneodesmosome interaction. At neutral pH (deep corneum) the rate of degradation of the corneodesmosomes by the KLK protease is hypothesised to be lower than that at low pH (superficial corneum) [21].

This subcellular system is disrupted in NS patients. The gene for the LEKTI inhibitor is mutated and, as a result, the concentration of LEKTI in the system is reduced. This is thought to be the reason NS patients have higher levels of KLK in the system, and is expected to cause an increase in the degradation rate of the cell's adhesion proteins [13].

## Multiscale models of the epidermis

In this paper we are interested in developing a multiscale model of the epidermis which incorporates a subcellular mass action model of desquamation into a multicellular overlapping spheres (OS) model of the tissue. There is a long history of OS models of the epidermis, beginning in 1995 with the model of Stekel et al. [22]. This early two dimensional model incorporated subcellular processes through an inverse square law which approximated the spread of signalling factors from cells. Over the following decades OS models have become more sophisticated, expanding to three dimensions and incorporating more specific subcellular information [23–26].

A common approach in multiscale OS models of epithelia is the inclusion of a subcellular ODE model for each cell [23–31]. For the epidermis, this approach has been used to investigate, for example, the role of TGF-$\beta$ (transforming growth factor $\beta$: a signalling molecule thought to help regulate division and differentiation) in wound healing using a mass action model [23, 24]. Another example is Kobayashi et al. [29] who used a phenomenological subcellular ODE model of calcium waves, from Kobayashi et al. [28], to understand calcium's role in epidermal homeostasis.

Another subcellular approach, also used to study the calcium gradient in the epidermis, is a molecular exchange model. Such a model incorporates the exchange of water and accompanying molecules, such as calcium, between adjoining cells. Grabe and Neuber [27] first implemented this in a two dimensional multiscale model and Sütterlin et al. [25] extended it to a three dimensional ellipsoidal model, to show such a system could maintain the calcium gradient itself and generate realistic tissue morphologies.

Three dimensional ellipsoid/spheroid, rather than spherical, models have been used by Sütterlin et al. [25] and Ohno et al. [26]. In these models, a cell changes its shape (flattens) as it enters a new layer of the tissue. Though such ellipsoidal models would be expected to produce more realistic packings and morphologies, there is a trade off with computational time. Additionally, these models do not account for cell rotation in their implementations, which one might expect to cause changes in packing, particularly at the interface between the different layers when there is shape discontinuity.

An alternative multiscale approach that has been used for the epidermis is a multicellular-extracellular model. For example, Schaller and Meyer-Hermann [32] included an extracellular reaction-diffusion model of nutrients and water to regulate division timing to investigate the persistence of melanoma cells in the epidermis under different parameter combinations [32].

A previously published multiscale model that incorporates subcellular dynamics for the adhesion degradation, and hence desquamation, is that of Ohno et al. [26]. The subcellular model is a phenomenological model to imitate the dynamics of the subcellular system described above. Desquamation of cells occurs when adhesion levels of the cell drops below a specified threshold (and has less than 16 cell neighbours). The authors include a deformable dermis layer in the model to understand the impact of the dermis on epidermis morphology. The study investigates homeostasis of the system, including tissue thickness, by modifying proliferation rates and stiffness of the dermis. This is in contrast to the results in this paper—we keep the proliferation rate constant and focus instead on changes to the desquamation process.

The difference between the desquamation model developed by Ohno et al. [26] and the model we propose in this paper is that Ohno et al. used a phenomenological model with parameters chosen by the authors. In contrast, our proposed model is mechanistic, using enzyme kinetics models, and fully parameterised using data from the experimental literature, allowing us to more deeply interrogate the effects of the molecular reactants. Additionally, we use upward force for the cell removal, to investigate how cells at the surface could interact with their environment, i.e. removal in squames, and the balance between the degradation rate of adhesion and experienced environmental forces. The importance of environmental force in desquamation was demonstrated by Goldschmidt and Kligman [33] who observed skin cell accumulation when participant's skin was covered by a cup for 6 weeks.

Three multiscale models that incorporate simpler (non ODE) models for adhesion degradation are Li et al. [34], Schaller and Meyer-Hermann [32], and Sütterlin et al. [25]. Both Schaller and Meyer-Hermann [32] and Sütterlin et al. [25] degrade the adhesion between cells according to the cell age and time since differentiation respectively. In both models, cells are removed using an adhesion threshold, making removal dynamics similar to methodologies based on cell removal at a specified age. Li et al. [34] add an upward force to the Schaller and Meyer-Hermann model [32] to pull cells from the tissue in order to separate and remove them, which is similar to the mechanics we implement (described in the Methods section). None of these three models investigate the molecular dynamics causing the adhesion degradation.

## Overview

In this study we incorporate a subcellular model of the desquamation process into our multicellular model from Miller et al. [35]. Previous models of the epidermis have incorporated degradation of cell-cell adhesion for desquamation using phenomenological models [25, 26, 32, 34]. However, no study has looked in detail at modelling the subcellular processes controlling adhesion degradation using a mechanistic approach. In this study, we implement the hypothesised subcellular biology of desquamation, described above and shown schematically in Fig 1, using a mass action model. The goal of the study is to determine whether the hypothesised model is sufficient to cause desquamation in the tissue and, in combination with a maintained proliferation rate, maintain a steady state tissue thickness. We show the results for this model at the subcellular scale for a single cell, highlighting the dynamics occurring as a cell migrates through the epidermis. We then investigate the desquamation process and epidermal homeostasis with a coupled subcellular-multicellular model, in both normal and abnormal (NS) scenarios, with the goal of determining how the tissue thickness is maintained, and how it responds to changes in inhibitor concentrations.

## Results

### Realistic adhesion degradation rates in the subcellular model require an effective concentration of enzyme

The subcellular model is developed for use in a multiscale model of the epidermis, however first we consider the subcellular response of a single cell moving through a pH gradient. The goal of this study is to determine whether the derived ODEs and parameters, calculated from experimental literature, result in a degradation rate of the adhesion proteins commensurate with the expected migration time of the cell. A mass action enzyme kinetics model, with competitive inhibition, is developed for the interactions between KLK, LEKTI, and corneodesmosome (CND), as shown in Fig 1, to predict the proportion of CND remaining for the cell. We define $[S]$, $[I]$, $[E]$, $[C_S]$, $[C_I]$, and $[P]$ as the concentrations of free substrate CND, free inhibitor LEKTI, free enzyme KLK, KLK-CND complex, KLK-LEKTI complex, and CND degradation products respectively. We can then define dimensionless variables $s = [S]/s_0$; $i = [I]/s_0$; $e = [E]/e_T$; $c_s = [C_S]/e_T$; $c_i = [C_I]/e_T$; and $p = [P]/s_0$ for some initial concentration of the substrate, $s_0$, and total amount of enzyme in the system, $e_T$. This gives the system of ODEs for the degradation of the adhesion as follows:

$$\frac{\mathrm{d}e}{\mathrm{d}t} = -k_{+1}s_0 es + (k_{-1} + k_2)c_s - k_{+3}s_0 ei + k_{-3}c_i, \tag{1}$$

$$\frac{\mathrm{d}s}{\mathrm{d}t} = -k_{+1}e_T es + k_{-1}\frac{e_T}{s_0}c_s, \tag{2}$$

$$\frac{\mathrm{d}i}{\mathrm{d}t} = -k_{+3}e_T ei + k_{-3}\frac{e_T}{s_0}c_i, \tag{3}$$

$$\frac{\mathrm{d}c_s}{\mathrm{d}t} = k_{+1}s_0 es - (k_{-1} + k_2)c_s, \tag{4}$$

$$\frac{\mathrm{d}c_i}{\mathrm{d}t} = k_{+3}s_0 ei - k_{-3}c_i, \tag{5}$$

**Table 1. The parameter values for the subcellular and multiscale model.** Values are given for the multiscale model and the rate parameters used for the single cell system are given in columns 3 and 5 respectively.

| Parameter | | Value: multiscale | | Value: single cell |
|---|---|---|---|---|
| KLK-CND Michaelis constant | $K_M$ | $4.60 \times 10^{-5}$ M | $K_M$ | $4.60 \times 10^{-5}$ M |
| $C_S$ degradation rate | $\hat{k}_2$ | $6.87 \times 10^3$ hr$^{-1}$ | $k_2$ | $2.29 \times 10^3$ hr$^{-1}$ |
| KLK-CND association rate | $\hat{k}_{+1}$ | $1.49 \times 10^8$ M$^{-1}$.hr$^{-1}$ | $k_{+1}$ | $4.97 \times 10^7$ M$^{-1}$.hr$^{-1}$ |
| Repulsive spring constant | $k$ | $150\,\mu$N [45] | | |
| Adhesion force shape parameter | $\gamma$ | 7 [34] | | |
| Normal adhesion force coefficient | $\alpha_0$ | $374.7\,\mu$N | | |
| Torsional spring constant | $k_\phi$ | $100\,\mu$N [35] | | |
| Cell cycle time | $T_C$ | $\sim U(13, 17)$ hr | | |
| Time step | d$t$ | 30 secs [38, 45] | | |
| Drag coefficient | $\eta$ | $0.1\,\mu$N.hr.$\mu$m$^{-1}$ [46] | | |
| Thickness calculation frequency | $f_h$ | 1 hr$^{-1}$ | | |

$$\frac{dp}{dt} = k_2 \frac{e_T}{s_0} c_s. \tag{6}$$

A full derivation of the model is given in the Methods section, and parameters are given in Table 1.

The model uses the cell's vertical location to determine its local pH. We define $\xi \in [0, 1]$ as the vertical dimension normalised by steady state corneum height with origin ($\xi = 0$) at the interface between the corneum and the lower epidermis (here we define the interface to be at $z = 4$). For any cells above the determined corneum height we set $\xi = 1$. The cell's local pH is then determined as a function of $\xi$:

$$\text{pH} = f_{\text{pH}}(\xi) = 6.8482 - 0.3765\,\xi - 5.1663\,\xi^2 + 3.1792\,\xi^3. \tag{7}$$

Using this, the rate parameters for the KLK-LEKTI interaction, $k_{+3}$ and $k_{-3}$, are determined:

$$k_{+3} = f_{+3}(\text{pH}) = (5.2\,\text{pH} - 19.5) \times 10^7 \; [\text{M}^{-1}.\text{hr}^{-1}], \tag{8}$$

$$k_{-3} = f_{-3}(\text{pH}) = 2.3 \times 10^6 \, e^{-3.0\text{pH}} \; [\text{hr}^{-1}]. \tag{9}$$

The proportion of remaining adhesion proteins (CND) is the output used by the multicellular system for the multiscale model. Further details on these functions are given in the Methods section, and in Section A in S1 Text.

We investigate the model dynamics for a single cell moving upwards through the corneum by assuming that the cell velocity is approximately constant through the corneum. This assumption is supported by results from our simulations where the variation in a cell's (daily) velocity as it migrates through the corneum is, on average, 5.7% (for the normal tissue described below). If we assume a migration time of 20 days [36] the approximate upwards velocity, normalised by corneum thickness, is $v_\xi = 0.05$ day$^{-1}$.

Results (shown in Fig E in S1 Text) showed the calculated system parameters, determined using *in vitro* and *in vivo* data from the literature, degraded the adhesion proteins significantly faster than expected: the corneodesmosome, $s$, degrades within a day ($s = 0.009$ at $t = 24$ hr),

rather than the expected time of 20 days (the migration time of the cell [36]). Degradation rates of $s$ over this 24 hour time period reach a maximum of 0.13 hr$^{-1}$ (at $t$ = 2 hr), despite the high level of enzyme that is in complex with the inhibitor: $c_i$ = 0.996 on average for $t \leq 24$ hours. These high rate values are due to the high effective rate parameter for complex formation in Eq (17): $k_{+1} e_T$ = −49.7 hr$^{-1}$. This indicates that either the rate parameters are wrong ($k_{+1}$ and $k_2$), or the mechanism is wrong.

We know many other processes and structures occur in the system that are neglected here which would affect the reaction. In particular, it has been hypothesised that tight junctions at the peripheries of the cells act as a barrier to the enzyme-corneodesmosome interaction at peripheral sites [16]. Similar effects may be occurring at the central regions of the cell, with the diffusion of the enzyme potentially obstructed by the intact corneodesmosomes themselves or the lipids that also reside in the extracellular space. Our results indicate that, assuming the determined rate parameters are of the right order of magnitude, this could have a significant effect on the degradation rate. By limiting the diffusion of the enzyme, the system is no longer homogeneously mixed and consequently the mass action model does not hold. A simple, and computationally efficient, way to counteract the effect of limited diffusion of the enzyme is to reduce its concentration, which reduces the effective rate parameter ($k_{+1}e_T$) in Eq (17) for degradation of $s$. The use of an *effective concentration of enzyme* is somewhat similar to using an effective volume fraction for crowded macromolecular diffusion [37]. For simplicity, we also keep $i_T = e_T$, as limited diffusion would have a similar affect on the inhibitor. A full investigation into the desquamation rate and the effect of the reactant concentrations is given in Section B in S1 Text.

Fig 2 shows the system dynamics with an effective concentration of $e_T$, reduced from 0.7 $\mu$m to 0.1 nm. The figure shows that the effective concentration is able to reproduce the rates expected for the desquamation process: the majority of $s$ degrades by T = 20 days, which is the expected migration time of a cell [36]. The dynamics of the system are strongly dependent the level of $c_i$ which determined by the ratio of $k_{+3}$ to $k_{-3}$, which varies with pH. More of the enzyme in complex with the inhibitor means there is less enzyme actively degrading the adhesion proteins ($s$), slowing the desquamation rate. These results support the hypothesis that the pH gradient in the skin is critical to controlling the rate of desquamation. This can also be seen in Fig D in S1 Text where we solve the system with constant pH levels.

## Incorporation of the subcellular model of adhesion into a multicellular model allows the study of desquamation

For our multiscale model of the epidermis we couple the ODE system for subcellular desquamation with a multicellular model of the tissue. This will allow us to investigate how events happening at a subcellular scale impact tissue dynamics—our aim is to determine whether subcellular reactions hypothesised in the literature can produce a homeostatic tissue thickness. Developing insights into how the scales interact is critical to developing a more comprehensive understanding of the tissue.

The multicellular model is an overlapping spheres model, with cell movement calculated using the inertia-less force balance:

$$\eta \frac{d\mathbf{c}_i}{dt} = \sum_{j \in N_i} \mathbf{F}_{ij} + \mathbf{F}_i^{\mathbf{Rt}} + \mathbf{F}_i^{\mathbf{D}} , \qquad (10)$$

where $\mathbf{c}_i$ is the cell centre location of cell $i$; $N_i$ is the set of neighbours of cell $i$; $\mathbf{F}_{ij}$ are the forces on cell $i$ due to interacting neighbour cell $j$; $\mathbf{F}_{i,\text{ext}}$ are any external forces acting on cell $i$; and $\eta$ is the viscous drag coefficient for the cell [38]. We define $\hat{\mathbf{r}}_{ij}$ as the unit vector between cells $i$

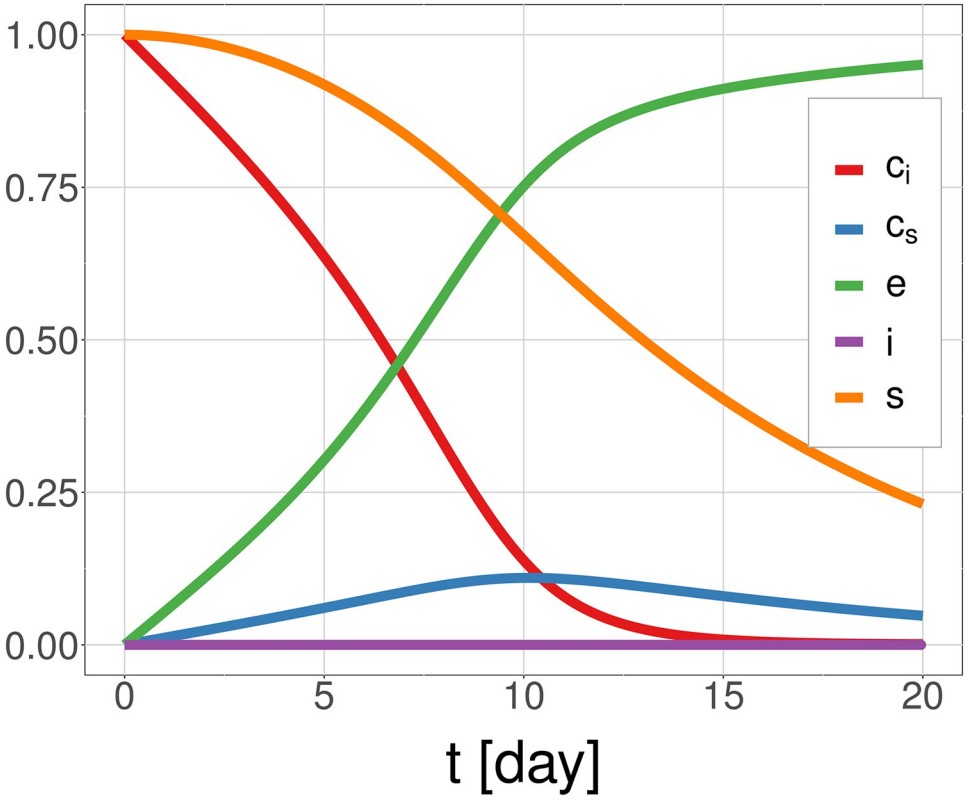

**Fig 2. Solution of the subcellular model for a cell moving at a specified velocity through the pH gradient of the corneum, using an effective concentration of KLK enzyme: $s_0 = 10 \; \mu M$, $e_T = i_T = 0.1$ nM.** This is a reduction in enzyme by a factor of $10^{-4}$ (from 0.7 $\mu M$).

and $j$ and $r_{ij} = \|\mathbf{c}_j - \mathbf{c}_i\| - R_{ij}$ where $R_{ij}$ is the sum of the cell radii (1 CD except when cells are dividing). The force of interest for the coupling between the models is the adhesion force between two cells, $\mathbf{F}_{ij}^{A}$ (this is $\mathbf{F}_{ij}$ for $r_{ij} > 0$, see Eq (31)). This force is based on the function used by Li et al. [34]:

$$\mathbf{F}_{ij}^{A} = -\alpha_{ij}\left((r_{ij}^* + c)e^{-\gamma(r_{ij}^* + c)^2} - ce^{-\gamma(r_{ij}^{*2} + c^2)}\right)\hat{\mathbf{r}}_{ij}, \; r_{ij} > 0, \quad \text{where } c = \sqrt{\frac{1}{2\gamma}}, \tag{11}$$

$$\text{and } r_{ij}^* = \frac{r_{ij}}{R_0}, \tag{12}$$

where $\alpha_{ij}$ is the adhesion coefficient between the two cells which is scaled by their CND level ($s$) from the subcellular model as described below, and $R_0 = 0.5$ cell diameters is the normal radius of a cell. A full description of all other forces in the model is given in the Methods section.

The cell's vertical location in the corneum in the multicellular system, i.e. $\xi$ (acting as a proxy for pH), provides the input into the subcellular model. For each cell, at each multicellular timestep, the subcellular model is integrated over the duration of the multicellular model time step to determine the cell's updated CND proportion ($s_i$). The new proportion of CND ($s_i$) is returned to the multicellular system and is used to scale the magnitude of the cell's

adhesion to its neighbours using the average of the two cell's CND levels:

$$\alpha_{ij} = \frac{1}{2}(s_i + s_j)\alpha_0 \,, \tag{13}$$

where $\alpha_{ij}$ is the adhesion coefficient in Eq (11) between cells $i$ and $j$ and $\alpha_0$ is the normal cell-cell adhesion coefficient. A diagram to illustrate this coupling is shown in Fig 3. As we explain in the Methods section, given the increased proliferation rate used in the model, the rate parameters for the multicellular system are scaled in order to reduce the height of the system for computational efficiency (Table 1).

The adhesion degradation alone is not sufficient to model desquamation. We also need to incorporate a 'removal force': a vertical force that is applied to 'surface cells', located in the top layer of the tissue, to pull the low adhesion cells upwards off the top of the tissue. Once these cells are sufficiently separated from the tissue they are removed from the simulation.

A more detailed description of the multicellular model, including the coupling and desquamation model, is provided in the Methods section. This model extends the (purely mechanical) model from Miller et al. [35] by incorporating the novel chemical subcellular model for adhesion degradation, and additional mechanics to model the resulting desquamation dynamics (as opposed to the sloughing model used in [35]).

Using this multiscale model we are able to simulate a tissue that balances the rate of proliferation and desquamation to produce a homeostatic tissue. Fig 4A shows a snapshot of a simulation, with cells coloured by their proportion of remaining CND, $s$. A video of this simulation is also provided in S1 Video. We can see the proportion of CND degrades homogeneously in the vertical direction and, at the top of the tissue, cells detach in clumps.

The outcome of interest for the multiscale model is the homeostatic corneum, and hence tissue, thickness. Fig 4B shows the mean thickness of the tissue over time and the estimated steady state thickness of the tissue and the corneum. The results shown summarise the

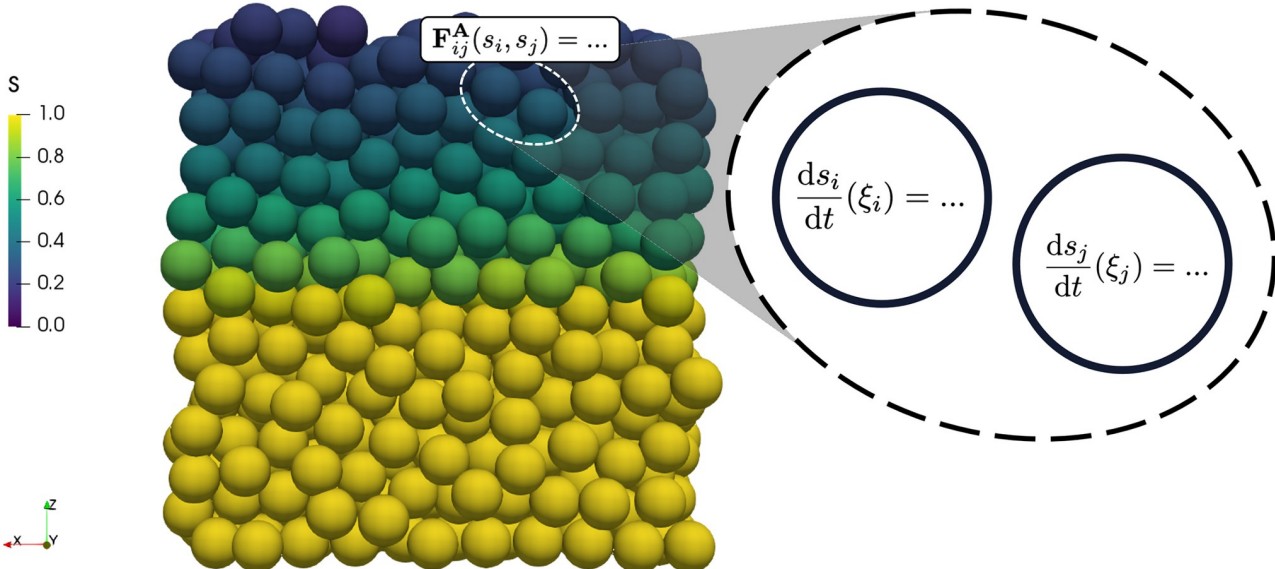

**Fig 3. Diagram of the coupling between the multicellular and the subcellular scales.** Cells are coloured by their proportion of remaining corneodesmosome. The cell gets its vertical location, normalised by corneum thickness and start height, ($\xi_i$) from the multiscale model, updates the subcellular model parameters and solves for that time step, and then returns the updated proportion of remaining corneodesmosomes ($s_i$) to the multicellular model to scale cell-cell adhesion ($\mathbf{F}_{ij}^{\mathbf{A}}$).

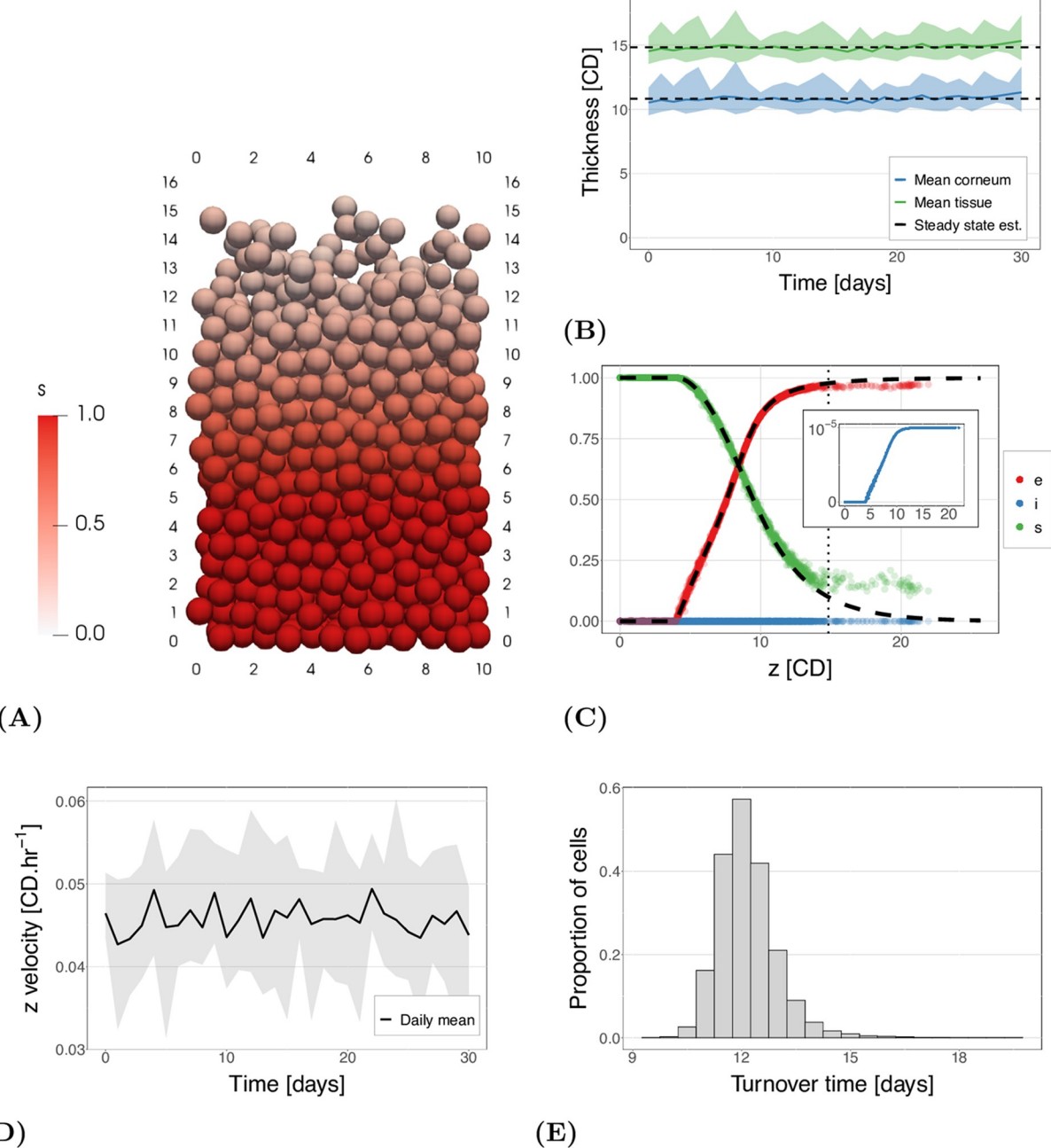

**Fig 4. Model results for normal homeostatic tissue.** (A) An example of the simulation output for one time point. Cells are coloured by their proportion of remaining corneodesmosome ($s$). A video of this simulation can be found in S1 Video. (B) The tissue thickness over time for 10 simulations. The dashed line shows the mean and the ribbon the minimum and maximum of all realisations over time. Green indicates the overall tissue thickness and blue the corneum thickness (4 CD less than the tissue). (C) Reactant levels for each cell at one time point in one simulation. The dashed black line shows the expected solution of the subcellular model given the expected vertical velocity. The inset shows the detail of the change in free inhibitor (LEKTI) levels as $i$ is defined as $i_T/s_0$ and therefore has a maximum possible value of $10^{-5}$ when $i_T = e_T = 0.1$ nM and $s_0 = 10\,\mu$M. (D) The average vertical velocity over time. The ribbon indicates the range seen across 10 simulations, and the line is the mean of the simulations. The velocity is calculated based on cells in the main tissue and does not include stem cells or cells that are experiencing the desquamation force. (E) Cell turnover time (age at which cells are removed from the system).

dynamics for 10 realisations of the system, to account for stochasticity in cell cycle times. We can see that this new model is able to maintain a steady state thickness. The steady state tissue and corneum thickness in Fig 4B is 14.8 CD and 10.8 CD respectively. We note there is a high variation in the height over time, which can also be seen in S1 Video. This is due to the way cells are removed: a cell/set of cells require a separation distance of 0.7 CD from the main tissue. As can be seen in Fig 4A, cells at the top of tissue can extend 2–3 CDs from the layer below, but still not be removed as a line of cells remains connected to the tissue (note the system is periodic in the horizontal directions). Consequently, when these cells detach, the tissue height decreases by 2–3 CD in one time step.

In order to better understand the maintenance of tissue thickness, and the interaction between the two scales, we analyse the dynamics of the subcellular model. The levels of the reactants over tissue thickness for one realisation, at $t$ = 30 days, are shown in Fig 4C. The point data shows the multicellular simulation values, while the lines indicate the expected values using the single cell solution. These expected values from the single cell are determined, as in Fig 2, using the mean velocity of cells in the multicellular system (shown in Fig 4D). This calculation only includes the 'migratory cells': cells that are not proliferative cells or surface cells (those experiencing the removal force).

Fig 4C shows little variation in the reactant concentrations, up until the steady state thickness (dotted line), in cells at any particular height ($s$ varies by ±3% on average). This can also qualitatively be observed in Fig 4A. Consequently, for a maintained proliferation rate, the tissue reaches a homeostatic state in which reactant levels are invariant in the horizontal directions and, given this, a cell's reactant levels at any location can be well-approximated by solving for a single migrating cell using the emergent cell velocity in the multicellular system. This causes a maintained homeostatic desquamation rate at the tissue level, and consequently a maintained tissue thickness.

Fig 4C also shows significant divergence in the levels above the steady state thickness (most significantly for $s$, the green points in Fig 4C). This divergence is an effect of the removal force applied to a cell when it is a surface cell in the top layer of the tissue. This force causes higher velocities compared to the velocities of migratory cells in the bulk of the tissue.

An important characteristic of the multiscale model, in relation to tissue thickness, is the level of CND at which cells separate from the rest of the tissue. Surface cells had a median substrate level of $s$ = 0.18. If we consider a tetrahedral tissue structure (i.e one cell is connected to three lower cells), $s$ = 0.18 is close to what is expected to be required to break the three cell-cell bonds to the lower cell layer for removal. Given that the removal force is half the maximum adhesion force, we would need $s \approx 0.17$ for desquamation for a cell experiencing the peak adhesion force with the lower three cells. Note, however, in reality the cell configuration is highly random (not tetrahedral) at the top of the tissue and, additionally, cells can 'drag along' cells below them when they are pulled from the top of the tissue.

Two aspects of the multicellular dynamics of the model relating to the cell proliferation and desquamation rate are the cell velocities and turnover times (time for a cell to traverse the tissue or corneum thickness). The cell velocity is determined by the rate of proliferation, and this, combined with the rate of cell loss, determines the tissue thickness. Cell turnover times provide the relationship between tissue thickness and cell velocity. The velocities and turnover times are shown in Fig 4.

Given a mean cell cycle time of 15 hours, and an expected layer thickness of 0.8 CD (for a tetrahedral packing), we expect cell velocities of 0.053 CD.hr$^{-1}$. The mean cell velocity from the timeseries data shown in Fig 4D is found to be slightly lower: $v_z$ = 0.046 CD.hr$^{-1}$, likely due to the random, rather than tetrahedral, packing. The mean velocity calculation excludes stem cells and surface cells (in the top layer of the tissue), as their velocities are much higher than in

the main tissue, and therefore their inclusion would bias the mean and not be representative of the main bulk of the tissue.

The cell turnover times are shown in Fig 4E. Given a corneum thickness of 10.8 CD and a velocity of $v_z = 0.046$ CD.hr$^{-1}$, we would expect a corneum turnover time of around 235 hours. As can be seen in the plot, the median cell age at death is 290 hours (12.1 days). The median age of cells at entry to the corneum is 82 hours, this gives a turnover time in the corneum of 208 hours—lower than expected. However, as noted above, the velocity does not include cells experiencing the removal force. The average velocity of cells experiencing the removal force is 0.31 CD.hr$^{-1}$, which would cause a decrease in the turnover time. Further reasons for this discrepancy are discussed in the results below for abnormal tissue (decreased LEKTI).

Experiments have shown evidence of two populations of proliferative cells in the basal layer with different cell cycle lengths [5–7]. We can investigate the effect of two proliferative populations in our model by comparing different combinations of fast and slow cycling proliferative cell populations with the same harmonic mean: $H(\text{T}_C) = 15$ hours. The harmonic mean ensures the overall proliferation rate is the same across each setup, as it accounts for the fact that fast cycling cells undergo more cycles than slow cycling cells. Due to this behaviour, we would expect an increased number of divisions to occur than would be inferred from the arithmetic mean. We also ran simulations using the same arithmetic mean which confirmed this.

To implement the two populations, we set 50% of the cells as fast cycling: $\text{T}_C = \text{T}_1$, and 50% as slow cycling: $\text{T}_C = \text{T}_1 + \Delta\text{T}_C$. Allocation of the cycle time was random—we did not account for any clustering of cells of the same proliferative type. Though this clustering has previously been observed in mouse epidermis [7], it would not be expected to have any significant effect on the thickness and average cell dynamics in the model. Simulation results and a detailed analysis are given in Section C in S1 Text. Results show increasing the cycle difference, $\Delta\text{T}_C$, but maintaining the same harmonic mean produces the same average proliferation rate and hence average $z$ velocities. Consequently, the steady state corneum thickness is also unchanged. From this study, we know we can approximate the expected steady state thickness of a system with two populations of proliferative cells by a single population system with cycle time equal to the harmonic mean of the two populations.

## Lower levels of LEKTI (inhibitor) cause premature desquamation

Netherton Syndrome (NS) is a disorder that mutates the gene for the LEKTI inhibitor, reducing LEKTI concentrations in the epidermis [13]. Presumably as a result of the reduced amount of inhibitor, KLK levels are elevated in NS patients: Komatsu et al. [13] recorded KLK levels in NS patients to be between 157% to 486% that of healthy corneum. In order to simulate the effects of NS we can extend our model to consider the effect of reduced LEKTI on the tissue by reducing the concentration of the inhibitor, $i_T$. The goal of this study is to determine how the tissue thickness is related to the proportion of available inhibitor in the model.

We define normal tissue as tissue where cells release normal levels of inhibitor, $i_T = e_T$ for all cells, and abnormal tissue as tissue where cells have reduced levels of inhibitor, $i_T < e_T$. Fig 5A shows the simulation output for abnormal tissue with no inhibitor present. Comparing this to Fig 4A shows no qualitative difference in the structure of the systems except the difference in corneum thickness. Fig 5B shows the mean corneum thickness for different levels of inhibitor. As would be expected, reducing the inhibitor reduces corneum thickness. This is because a reduced level of inhibitor increases the enzyme available to degrade the adhesion proteins, resulting in earlier desquamation. We can see this in Fig 5C, where decreasing concentrations of inhibitor increases the amount of free enzyme $e$, and this increases the degradation rate of the adhesion protein $s$. We note that Fig 5C shows clearly the initial conditions for each

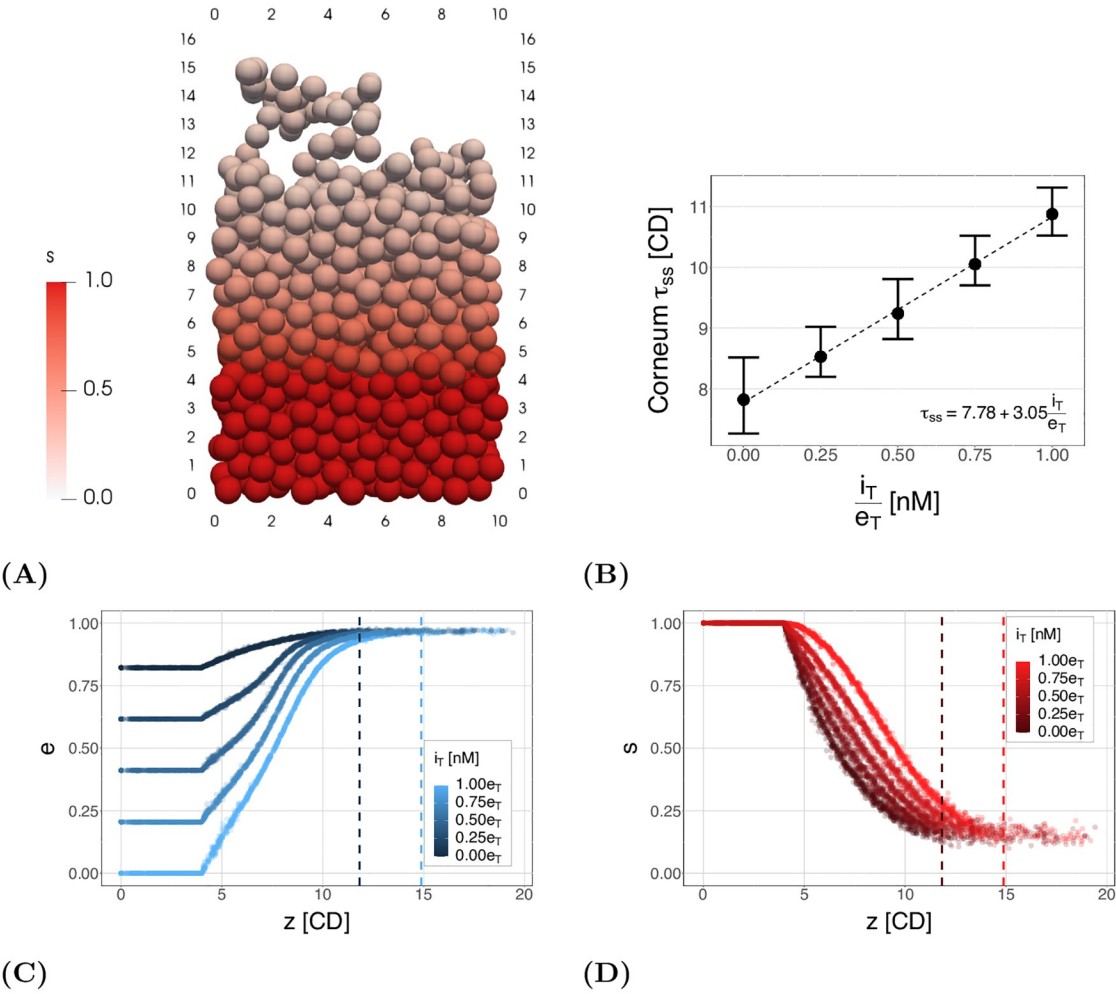

**Fig 5. Tissue structure and thickness results for different levels of inhibitor.** (A) Simulation output from one simulation (with no inhibitor) at one time point, showing the level of adhesion protein *s* in each cell. Note, there is no apparent difference in system dynamics compared to the normal system, just an increased rate of reduced *s* in time and consequently seen in the vertical direction in the image. A video of this simulation is provided in S2 Video. (B) The steady state thickness values. The points show the mean and the bars indicate the minimum and maximum thickness over the 10 realisations of each setup. The dashed line is a linear fit to the points, with the equation given in the lower right. (C) and (D) Abnormal tissue *s* and *e* levels with different levels of inhibitor. The data is taken from the final time point of one simulation for each inhibitor concentration. The dashed lines show the steady state heights for $i_T = 0$ (black) and $i_T = e_T$ (blue or red).

inhibitor level at entry to the corneum: we assume all inhibitor (*i*) is in complex with the enzyme (*e*), and any remaining enzyme is free enzyme, as described later in the Model section.

Fig 5B also shows that corneum thickness, $\tau_{ss}$, is linearly dependent on the concentration of inhibitor present in the system. A linear fit is given on the plot. When there is no inhibitor present the corneum thickness decreases by 28%. Though we have no experimental data for reduction in corneum thickness for NS patients, we can make a comparison with data from Komatsu et al. [13] using the amount of free KLK in the system. The study found that NS patients had KLK levels 157%–486% that of healthy corneum. We can calculate the amount of free enzyme in the system by taking the sum of the level of *e* for all cells in the corneum, up to the steady state thickness (we use the average over time). Comparing this for the normal and abnormal model results (Figs 4C and 5C), the abnormal tissue with no inhibitor has 190% the

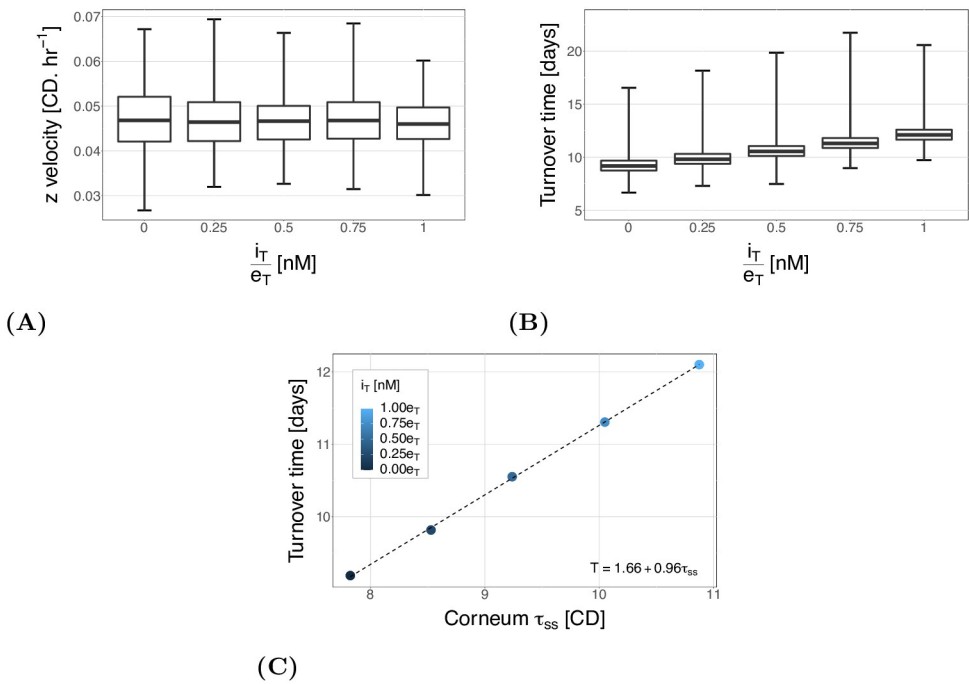

**(A)**                                        **(B)**

**(C)**

**Fig 6.** (A) The mean vertical cell velocity for different levels of inhibitor. The box plot shows the quartiles, and the error bars give the minimum and maximum across all cells and simulations. The velocity profile is similar across all setups. (B) The cell turnover time, from birth to desquamation, for different levels of inhibitor. The box plot shows the quartiles, and the error bars give the minimum and maximum across all cells and simulations. The box plot shows there are a large number of outlier cells with high turnover times. (C) The steady state against the cell turnover time. The dashed line is a linear fit to the points, with the equation given in the lower right.

free enzyme of the normal system—within the bounds of the observed increase, but towards the lower end despite being the worst case scenario. In Section B.4 in S1 Text we also find the use of an effective enzyme concentration decreases the effect of a reduction in inhibitor on total free enzyme when solving the subcellular model for a single migrating cell. Consequently, the 28% thickness reduction is expected to be a lower bound for the effect of the reduced inhibitor on tissue thickness.

We can also look at the change in the vertical velocity and turnover time. Changes to the inhibitor concentration should have no effect on the proliferation, and so we would not expect the velocity of the cells to change. This can be seen in Fig 6A, where there is negligible difference in mean velocity (relative range for the means is 2.5%).

Fig 6B shows that the turnover time increases with the decreased inhibitor. This is to be expected as, given the same velocity, an increased thickness for the cell to traverse would cause an increase in migration time. This can be seen clearly if we directly compare the turnover time and steady state corneum thickness (Fig 6C). The figure shows turnover time is proportional to corneum height, as expected, with a linear fit given on the plot.

If we consider the linear relationship between turnover time and thickness shown on Fig 6C, we could expect the coefficient of $\tau_{ss}$ to be 0.91 days.CD$^{-1}$ (rather than 0.96) given an average vertical velocity of $v_z = 0.046$ CD.hr$^{-1}$, and the constant to be the average age at entry to the corneum: 3.4 days (rather than 1.66). There are three factors that are likely to contribute to these discrepancies. Firstly, the calculated turnover time is not identical to the average cell age at the steady state tissue thickness. This is because tissue height is determined by the mean

height of the top layer of cells, which is lower than the mean height at which cells are removed, increasing the coefficient and intercept of the fit. Secondly, the velocity is calculated using only 'normal cells': cells which are not stem or surface cells. This is because the forces on the surface cells (and the stem cells), are much stronger and therefore including these cells is not a good representation of the bulk tissue velocity. Consequently, the period of time in which cells are a surface cell, and moving at higher velocities than the average, would be expected to reduce both the coefficient and intercept. Thirdly, there are only 10 data points (simulations) for each steady state height calculated and so stochasticity in the system will have an effect.

## Heterogeneous full recovery is more effective than homogeneous partial recovery

In NS, no treatments currently exist that address the cause of the disease, only the symptoms. One example of a therapy that may affect the system considered here is (narrowband ultraviolet B) phototherapy. In the treatment of psoriasis the same therapy has been observed to induce apoptosis, growth arrest, and possibly allow for DNA repair [39]. In NS patient case studies the treatment has been reported to cause 'clinical' or 'marked' improvement for the patients [40, 41]. Consequently, it is possible this treatment results in only partial recovery of normal function in the tissue.

Partial recovery after treatment, rather than full recovery of normal function, is a scenario that can generally occur in many disease treatments. If we consider a treatment that attempts to restore function at a cellular or subcellular level, it is possible to hypothesise two scenarios which could result in only a partial recovery: partial restoration of normal function in all cells (homogeneous recovery), or restoration of normal function in only a subset of cells (heterogeneous recovery). If we consider the subcellular system described in this paper, the first scenario would be an increase in inhibitor levels across all cells. In the second scenario, only a proportion of cells would recover normal inhibitor levels and remainder would still have reduced inhibitor levels. It is also probable that the response would be a combination of the two scenarios: different levels of improvement in different subsets of cells.

In this section we compare the two limiting scenarios: heterogeneous and homogeneous recovery, for a hypothetical treatment that restores some level of inhibitor to cells in the system. The efficacy of homogeneous recovery can be seen above in Fig 5B where results show a linear relationship between the inhibitor concentration in the cells and the tissue thickness. For heterogeneous recovery, we assume that the level of inhibitor in any cell is inherited from its parent stem cell and assign a level of inhibitor to each stem cell. The allocation of the abnormal and normal inhibitor levels to cells is random, so there is no spatial dependency. We set $i_T$ = 0 for the abnormal cells and $i_T = e_T$ for the normal cells. Note there are 100 stem cells in the system. The proportion of normal to abnormal stem cells persists throughout the simulation given the cell lineage used (a stem cell divides to create one stem and one differentiated daughter).

Fig 7A shows the thickness of the tissue with different proportions of abnormal cells. As can be seen in the figure, increasing the number of normal stem cells has a non-linear effect on the corneum thickness. Here, we have fitted a quadratic curve to the thickness (shown on the figure). Fig 7A also includes a line to indicate what might be expected if the improvement in tissue thickness was linear, as in Fig 5B. Consequently, if we consider the improvement in each system—heterogeneous compared to the homogeneous recovery shown by Fig 5B—in terms of the 'total recovered inhibitor' due to treatment, better outcomes are seen if a proportion $p$ of cells recover full inhibitor concentration, compared to recovering the same proportion $p$ of the inhibitor in all the cells.

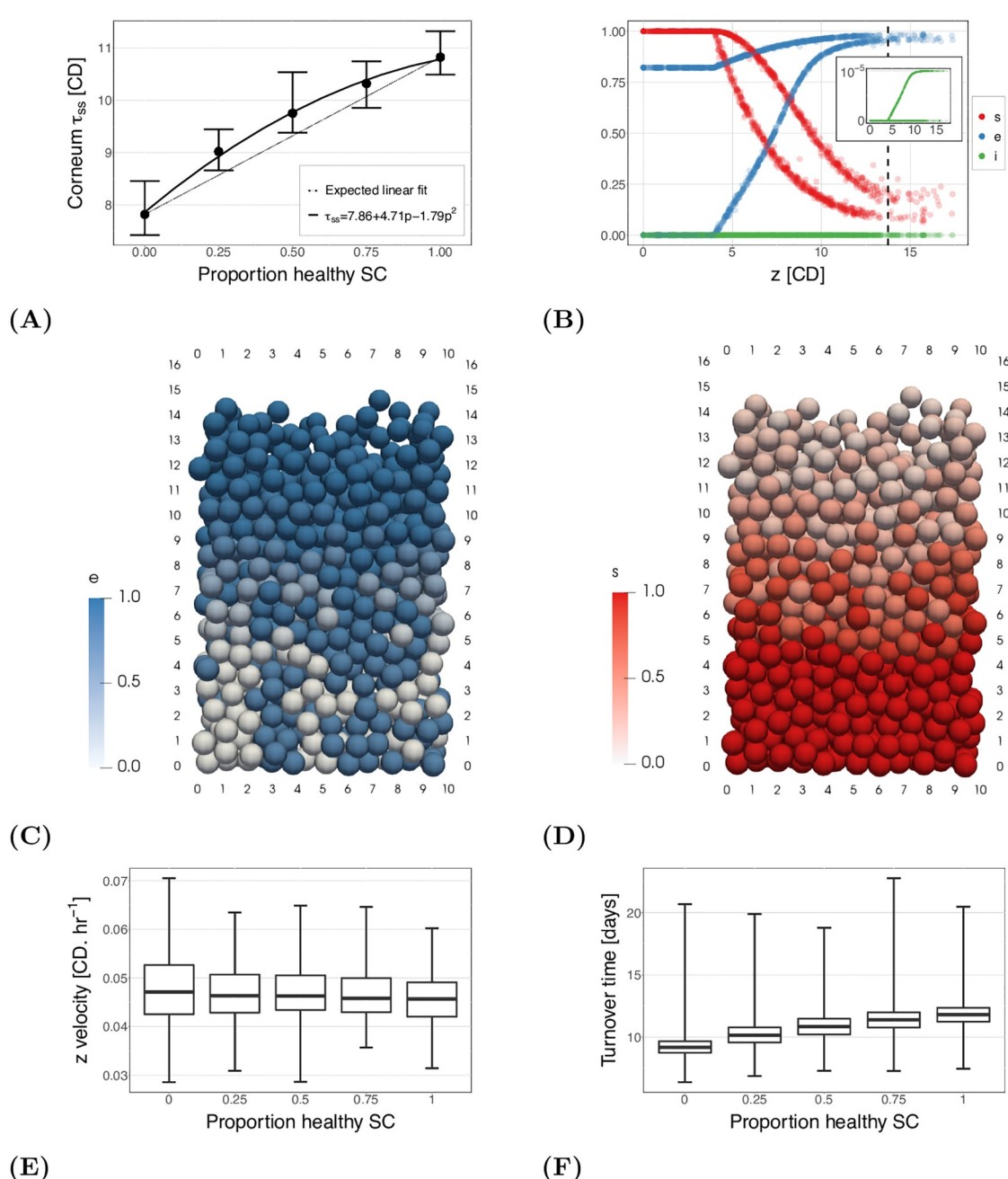

**Fig 7. Simulation results for different proportions of normal stem cells (generate cells with normal inhibitor levels) to abnormal stem cells (generate cells with no inhibitor).** (A) Steady state corneum height. The points are the mean and the bars indicate the minimum to maximums for the 10 realisations of each setup. The solid line is a quadratic fit (equation given in legend), and the dotted line shows the expected results for a linear relationship. (B) Reactant levels of each cell at one time point with 50% normal stem cells and 50% abnormal (no) stem cells. The dashed line is the steady state height. The two paths for both $s$ (red) and $e$ (blue) are the different subcellular system dynamics for cells that have normal inhibitor levels (lower $e$ and slower reduction in $s$) compared to cells with no inhibitor (higher $e$ and faster reduction in $s$). (C) and (D) Simulation snapshots with cells coloured by their proportion of free enzyme, $e$ (C) or proportion of remaining substrate, $s$ (D). A video of this simulation can be found in S3 Video. (E) Cell velocities. The box plot shows the quartiles, and the error bars give the minimum and maximum across all cells and simulations. The velocity profile is similar across the different proportions. (F) Turnover times of differentiated cells. The box plot shows the quartiles, and the error bars give the minimum and maximum across all cells and simulations.

Fig 7B shows the reactant levels of the cells at one time point in a simulation with half the stem cells abnormal and the other half restored to normal inhibitor levels. As would be expected, half the cells (points) follow the abnormal path and the other half the normal path. Fig 7C and 7D are simulation snapshots with cells coloured by the $e$ and $s$ levels (a video can also be found in S3 Video). The abnormal cells can be identified as the white cells when $z < 4$ CD in Fig 7C, and the normal cells are blue. This shows their initial conditions which are assigned at birth, based on the proportion of $i_T$ to $e_T$.

The plots and simulation snapshots in Fig 7 show that the abnormal cells are not lost before the normal ones, as might be expected given their increased adhesion degradation rate. As cells tend to be removed in clumps the abnormal cells are held longer in the tissue, or in the clumps, by the normal cells.

Finally, Fig 7E and 7F show the cell velocities and turnover times against the proportion of recovered cells. Similar results are seen to the abnormal system: the mean velocity is similar between setups, as the proliferation rate is not changed (in this case it varies by 3.7%); and turnover time follows the same functional shape as corneum thickness. We find the same linear relationship between the steady state height and the median turnover time seen in Fig 6C (results not shown for brevity), as expected given the velocity does not change. If we calculate the fit: coefficient 0.98 days.CD$^{-1}$ and constant 1.5 days, it is not exactly the same as found in Fig 6C. The slight variation in the coefficient and the constant is likely explained by stochasticity in the system.

## Discussion

### Emulating a homeostatic tissue

Our new subcellular model for desquamation supports the hypothesis of desquamation driven by KLK enzymes. In this hypothesis KLK enzymes degrade the corneodesmosomes (adhesion proteins) between the cells, a process which is inhibited by LEKTI and regulated by local pH. Though the results show that enzyme dynamics follow the expected function over pH, they also reveal the model is missing some components of the system—the degradation occurs too quickly to match observed desquamation times *in vivo*. These missing components likely include the presence of the corneodesmosomes and other extracellular complexes limiting the diffusion of the enzymes, and therefore reducing the rate of degradation of the corneodesmosomes; and extrapolation of *in vitro* data to an *in vivo* environment. Macromolecular crowding is a known phenomenon in biological media generally, and modelling the resulting diffusion dynamics is a very complex problem [37, 42]. In order to compensate for this limited diffusion effect, without incorporating complex new processes in the model, we instead propose the use of an *effective concentration of enzyme*. Limited diffusion would reduce the amount of available enzyme to bind to the substrate. Decreasing $e_T$ has the same effect as it directly affects the effective rate parameter for the degradation of $s$ ($k_{+1} e_T$). Consequently this is a way to simulate limited diffusion in a computationally efficient way. We maintain $i_T = e_T$, as limited diffusion would also affect the inhibitor. Fig 2 shows a reduced concentration of $e_T$ is able to reproduce the rates expected for the desquamation process (on the scale of 20 days [36]).

In this paper we have introduced a multiscale model for desquamation of the epidermis. The coupling of the two scales allows us to investigate whether the subcellular model is sufficient to maintain desquamation at a tissue level, and how the tissue responds to changes to subcellular parameters. In Fig 4 we showed that the coupled model is able to maintain a homeostatic tissue height. The height of the tissue is determined by the balance between the proliferation rate and the desquamation rate. The system with reduced LEKTI in the subcellular model (abnormal tissue) has an increased desquamation rate, changing this balance and

hence the thickness of the tissue. Consequently, we can use such a multiscale system to investigate the effect of different aspects of proliferation and desquamation on tissue thickness.

As a simplified alternative, from Fig 4C, we know all cells in the system follow a similar reduction in *s* (adhesion proteins). Therefore it would be possible to approximate the decay with a sigmoidal (or equivalent) function. However, this would not provide the same level of information of the interaction between the subcellular and multicellular dynamics as the full multiscale model.

## Insights into the system

We have used our multiscale model to investigate the effect of different epidermal attributes on tissue height and dynamics. The first attribute is the existence of two proliferative cell populations. Using the model we showed that the dynamics of a tissue with two proliferative populations can be approximated by a tissue with a single proliferative population. The proliferation rate of this single population needs to be equal to the harmonic mean of the two proliferative populations.

The second investigation analysed an abnormal tissue: the genetic disease Netherton Syndrome, known to reduce production of the LEKTI inhibitor in the desquamation process. Our subcellular model results showed a reduction in inhibitor is sufficient to reproduce the increased concentration of free enzyme observed in patient skin samples. Additionally, the multiscale model showed a decrease in the inhibitor has a noticeable effect on tissue thickness, though model approximations cause the system to underestimate the effect of the reduced inhibitor. The relationship between the inhibitor concentration and tissue thickness is also shown to be linear. If we considered a hypothetical treatment for the abnormal cells, that only cures a proportion of the proliferative cells, tissue thickness increases approximately quadratically with the number of cells treated. Consequently, full cure of a proportion of proliferative cells is shown to be better than partial cure of all proliferative cells (Fig 7A) for an equivalent recovery of total inhibitor in the system.

## Limitations and future work

Several simplifications are always necessary in order to produce a computationally efficient system. In addition, there are likely a multitude of processes occurring at both a subcellular and multicellular level that are unknown and affect the processes described by this model.

With respect to the subcellular model, one known limitation is not explicitly accounting for the structure of the extracellular space in degradation of the adhesion structures. As discussed in the results, it is possible that the adhesion structures we consider here (corneodesmosomes) are limiting the diffusion of the enzyme, as well as other structures, such as tight junctions, which are not included in the model [16]. Future work could look in more detail at the potential affect of limited diffusion on the degradation rate. However, this study would currently be limited by a lack of data for parameterisation.

Related to this, the parameterisation of the subcellular model is built around data that is often unable to be recorded *in vivo*. Consequently, it has been necessary to extrapolate much of the data from a laboratory setting to a physiological setting. This applies to both the rate parameters and reactant concentrations. Such an extrapolation requires many assumptions which are unable to be tested. Consequently, as potential improved testing methodologies, or more comprehensive data, become available it will be possible to improve this model. We are also interested in performing a parameter sensitivity analysis on the full system, for the subcellular parameters, extending the work presented in Section B in S1 Text, to better understand the sensitivity of the model results to each of the rate parameters and reactant concentrations.

Another area that is affected by a lack of data is the force experienced by the cells to enable desquamation. Though it appears they are important for desquamation [33], it is difficult to understand the forces applied to the skin in everyday life, both in type (i.e. shear or tension) and magnitude. However, given we currently ignore any anisotropy in the distribution and degradation of the adhesion molecules, and additionally do not know the strength of the adhesion molecules between cells, this level of detail is not yet useful. We are interested in investigating the anisotropy of degradation in future work. We expect such a study would be strongly linked to the proposed study above on the structure of the extracellular space and limited diffusion.

Two further simplifications of the multicellular model are the use of spherical, rather than ellipsoidal, cells and a flat basal membrane. In the stratum corneum, cells are almost flat disks with a diameter of 20–40 $\mu$m, and a height of less than 0.5 $\mu$m [8–10]. The use of spherical cells will clearly influence these results, as the packing of the cells will change depending on cell shape. This would likely scale the determined function linking subcellular parameters and tissue height, however we would not expect it to change the shape of these functions and hence the discussed insights from these results. With the use of an undulating rather than a flat membrane, we would expect much more variation in the results with respect to turnover time and velocities. Previous studies have also shown that the undulation of the membrane could have an impact on the tissue thickness [26, 43]. However, we would not expect it to qualitatively change the conclusions of the results.

## Methods

In this section we detail the implementation of our multiscale model. First, we describe the derivation of the subcellular model for adhesion degradation, including the ODE system, parameterisation, and initial conditions. Then we briefly present the base multicellular model, and the coupling between the subcellular and multicellular models. Finally, we detail how we implement force based desquamation in the multiscale model.

### A subcellular system

We are interested in developing a subcellular model of the adhesion degradation over time for the process shown in Fig 1. This model takes a spatial pH gradient as an input, and then models the interaction between the KLK enzymes, LEKTI inhibitor, and adhesion proteins: corneodesmosomes (CNDs) for each cell individually using mass action kinetics. The rate parameters depend on the changing pH as the cell moves up through the tissue and experiences the pH gradient. The output of this model is the proportion of CND remaining for each cell, which scales the cell-cell adhesive force of the multicellular model.

**Subcellular differential equations.**  We convert the system described in Fig 1 into the competitive inhibition chemical equations:

$$E + S \underset{k_{-1}}{\overset{k_{+1}}{\rightleftharpoons}} C_S \overset{k_2}{\rightarrow} E + P , \tag{14}$$

$$E + I \underset{k_{-3}}{\overset{k_{+3}}{\rightleftharpoons}} C_I . \tag{15}$$

where $E$ is the KLK enzyme, $S$ is CND, $I$ is the inhibitor LEKTI, P is the product(s) corneodesmosome degradation produces, $C_S$ is the complex formed between the KLKs and the corneodesmosome, and $C_I$ is the complex formed between the KLKs and LEKTI. All rate parameters, $k_{+1}$, $k_{-1}$, $k_2$, $k_{+3}$, and $k_{-3}$, are initially assumed to be functions of local pH. We also assume

there is no degradation of the inhibitor or enzyme over the time scale of the cell's migration time through the corneum (approximately 20 days in the epidermis [36]).

We take this system of chemical equations and produce a set of ODEs using mass action kinetics. In order to simplify the computation and analysis, we scale the system by the enzyme and corneodesmosome concentrations. We define the following dimensionless variables: $s = [S]/s_0$, $i = [I]/s_0$, $p = [P]/s_0$, $e = [E]/e_T$, $c_s = [C_S]/e_T$, and $c_i = [C_I]/e_T$, where $[X]$ is the concentration of species X in Eqs (14) and (15); $s_0$ is the initial concentration of adhesion proteins; and $e_T = [E] + [C_S] + [C_I]$ is the total enzyme present, which is conserved. This gives the following system of equations:

$$\frac{\mathrm{d}e}{\mathrm{d}t} = -k_{+1}s_0 es + (k_{-1} + k_2)c_s - k_{+3}s_0 ei + k_{-3}c_i \,, \tag{16}$$

$$\frac{\mathrm{d}s}{\mathrm{d}t} = -k_{+1}e_T es + k_{-1}\frac{e_T}{s_0}c_s \,, \tag{17}$$

$$\frac{\mathrm{d}i}{\mathrm{d}t} = -k_{+3}e_T ei + k_{-3}\frac{e_T}{s_0}c_i \,, \tag{18}$$

$$\frac{\mathrm{d}c_s}{\mathrm{d}t} = k_{+1}s_0 es - (k_{-1} + k_2)c_s \,, \tag{19}$$

$$\frac{\mathrm{d}c_i}{\mathrm{d}t} = k_{+3}s_0 ei - k_{-3}c_i \,, \tag{20}$$

$$\frac{\mathrm{d}p}{\mathrm{d}t} = k_2\frac{e_T}{s_0}c_s \,. \tag{21}$$

**pH gradient.**   The local pH is the input for the ODE system for the cell, and depends on cell location. The pH gradient was fit to data obtained from a graph in Ohman et al. [21] for pH in human forearm, abdomen, and calf epidermis collected using sello-tape. The function fit for the pH gradient, with an R-squared of $R^2 = 0.94$, is as follows:

$$\mathrm{pH} = f_{\mathrm{pH}}(\xi) = 6.8482 - 0.3765\,\xi - 5.1663\,\xi^2 + 3.1792\,\xi^3 \,, \tag{22}$$

where $\xi \in [0, 1]$ is the height of the cell above the base of the corneum as a proportion of the corneum thickness. For any cells above the determined steady state thickness we set $\xi = 1$.

**Rate parameters for the subcellular system alone.**   Eqs (16) to (21) require five rate parameters: three for the interaction between the KLK enzyme and the corneodesmosomes; $k_{+1}$, $k_{-1}$, and $k_2$, and two for the interaction between the KLK enzyme and LEKTI inhibitor; $k_3$ and $k_{-3}$. We can estimate values for these parameters, the pH gradient, and the reactant concentrations using data from the literature. The parameters determined here are for the subcellular model results. These parameters are modified for the multiscale model implementation, which is discussed in the Multiscale model section. We provide a summary here, for more details see Section A in S1 Text.

The rate parameters for the LEKTI-KLK (inhbitor-enzyme) reaction, Chemical Eq (15), are determined from *in vitro* experiments by Deraison et al. [12]. This study determined association and dissociation rates between LEKTI and KLK molecules at 4 different pH values. A linear regression (using a log transform for $k_{-3}$) on this data produces the following equations for

$k_{+3}$ and $k_{-3}$ as functions of pH:

$$k_{+3} = f_{+3}(\text{pH}) = (5.2\,\text{pH} - 19.5) \times 10^7 \; [\text{M}^{-1}.\text{hr}^{-1}], \tag{23}$$

$$k_{-3} = f_{-3}(\text{pH}) = 2.3 \times 10^6 \, e^{-3.0\text{pH}} \; [\text{hr}^{-1}], \tag{24}$$

with R-squared values of $R^2 = 0.94$ and $R^2 = 0.9998$ respectively. These fits are plotted in Fig C in S1 Text.

The data used to parameterise the CND-KLK (adhesion-enzyme) interaction, Chemical Eq (14), is from Caubet et al. [18]. This study measured the degradation of two corneodesmosome proteins incubated with KLK5 in both neutral (pH = 5.6) and acidic (pH = 7.2) pH over a two hour incubation period. However, as the authors point out, there appears to be little variation between the two pH levels in the response. Additionally, this data set is small and so it is unreliable for calculating fits with high confidence when considering the different pH levels separately. Consequently, for this reaction we assume the effect of pH is negligible.

In order to determine the parameters to the data from Caubet et al. [18] it is necessary to simplify the system. Given the experiments were performed without inhibition, we can reduce the set of Eqs (16) to (21) and apply the quasi-steady state assumption of Briggs et al. [44]. This gives a relationship between remaining $s$, and time which can be solved to determine $k_2$ and the Michaelis constant, $K_M$:

$$K_M = \frac{k_{-1} + k_2}{k_{+1}}. \tag{25}$$

A full description of the relationship and the fits is given in Section A in S1 Text. The calculated values are shown in Table 1.

From the Michaelis constant, $K_M$, it is necessary to determine the values of $k_{+1}$ and $k_{-1}$ in order to solve the system. Given $K_M$ is on the order of $10^{-5}$ and $k_2$ on the order of $10^3$, $k_{+1}$ is approximately five to eight orders of magnitude greater than $k_{-1}$ (from Eq (25)). Consequently, we assume that the value of $k_{-1}$ is negligible and we approximate $k_{-1} = 0$. Solving the full ODE system for different values of $k_{-1}$ at both acidic and neutral pH supports this assumption—the change in $s(T)$ is less than 0.01% at either pH for $k_{-1} \in [0, 10^4]$. Consequently, we get the the value for $k_{+1}$ shown in Table 1.

**Concentrations of reactants.** There is limited *in vivo* data for the concentrations of KLK, LEKTI, and CND in the corneum. Most data in the literature is not available as concentrations, as is required for the model, and hence has to be approximated. Consequently, these values are very rough estimations, and only provide an idea of the order of magnitude of the relative concentrations of the reactants. A summary of how the concentrations were determined is given below, with more detail in Section A in S1 Text.

Several studies have investigated the weight of KLK in epidermal tissue. The data is commonly provided as a weight of free enzyme per weight of dry corneum tissue. Consequently, it does not account for enzyme in complex and provides no spatial component to the concentrations. It is necessary, for the purposes of the model, to convert these amounts to molar concentrations in extracellular space. This was done using an approximate molecular weight of KLK5, the water content of the corneum, and the volume ratio of cell to extracellular space in the corneum. As a simplification, due to limited data, we assume that the volume of the cell and the extracellular space is constant throughout the corneum. Values for all of these parameters are given in Table 2.

**Table 2. Parameters used to determine the concentration of KLK5 enzyme.**

| Parameter | Value | Reference |
|---|---|---|
| Dry weight of KLK5 enzyme | 3.1 ng.(mg dry tissue $^{-1}$) | [13] |
| Molecular weight of KLK5 | 33 kDa | [20] |
| Water content of corneum | 0.5 g.g$^{-1}$ | [47] |
| Volume proportion of extracellular water | 13% | [8, 48] |
| Estimated enzyme concentration | 0.723 $\mu$M | |

**Table 3. Parameters used to determine the concentration of corneodesmosome proteins.**

| Parameter | Value | Reference |
|---|---|---|
| Protein count on periphery of cell | 16 $\mu$m$^{-1}$ | [16] |
| Protein count on central region of cell | 10 $\mu$m$^{-1}$ | [16] |
| Dimensions of cell (width × height) | 30 × 0.3 $\mu$m | [8] |
| Extracellular space between cells | 0.044 $\mu$m | [48] |
| Estimated corneodesmosome protein concentration | 6.6 $\mu$M | |

The second reactant is the inhibitor LEKTI. A study by Fortugno et al. [20] determined LEKTI fragments most effective at inhibiting KLK5 were present in the same molar quantities as KLK5. Consequently, we set $i_T = e_T$ in normal epidermis.

The final reactant is the corneodesmosomes (adhesion proteins). The *in vivo* data on corneodesmosomes provides counts of associated proteins across the edges of cells. In order to convert this into a concentration we need to determine the number of proteins per unit of extracellular volume. This requires information on the cell size and the ratio of cells to extracellular space. This is given in the Table 3. We note the corneodesmosome concentration is only one order of magnitude greater than the enzyme concentration.

**Initial conditions.** The initial conditions for the subcellular model are the concentrations of the reactants upon entry to the corneum (assigned at birth in the model). Corneodesmosomes are at their maximum concentration initially ($s(t = 0) = 1$) as the cells have the maximum adhesion protein concentration at the start of the corneum. We assume the enzyme and inhibitor start in complex ($c_i$), limited by whichever of the enzyme or inhibitor has a lower concentration. If there is more enzyme than inhibitor, we could assume that the remaining enzyme is free, i.e. $c_s(t = 0) = 0$, however this is a stiff system and the solution therefore requires very small time steps to solve and has the potential to cause numerical issues (as can be seen in Section B in S1 Text). Consequently, we instead assume that any enzyme that is not in complex with the inhibitor is in an instantaneous quasi-equilibrium with the substrate-enzyme complex.

In order to calculate what the equilibrium state would be we first assume all inhibitor is in complex with the enzyme. For the remaining enzyme, we assume the enzyme-substrate reaction, Eq (16), is approximately equal to zero. Given we are estimating the dynamics instantaneously upon the release of these reactants, we set $s = 1$. This gives the following relationship between $e$ and $c_s$:

$$-k_{+1}s_0 e + (k_{-1} + k_2)c_s = 0 . \tag{26}$$

We know $k_{-1} = 0$ and, additionally, due to conservation of the enzyme $e + c_s + c_i = 1$. Therefore the initial conditions become:

$$e(t = 0) \quad = e^* = \frac{k_2}{k_{+1}s_0 + k_2}\left(1 - \frac{i_T}{e_T}\right), \tag{27}$$

$$c_s(t = 0) \quad = 1 - e^* - \frac{i_t}{e_t}, \tag{28}$$

$$c_i(t = 0) \quad = \frac{i_T}{e_T}, \tag{29}$$

with $s(t = 0) = 1$ and $i(t = 0) = 0$. Parameters $i_T$ and $e_T$ are the concentration of inhibitor and enzyme respectively, with $i_T \leq e_T$.

## Multiscale model

In order to build the multiscale model we incorporate the ODE system described in Eqs (16) to (21) into our multicellular model. This ODE system regulates the degradation of adhesion for each cell so that it can undergo the desquamation process. In order to efficiently incorporate the subcellular model described above we scale the rate parameters to suit the parameters of the multicellular model and speed up computation. In order to effectively model the desquamation process using adhesion degradation, we also incorporate a removal force into the multicellular model so that we can determine the cells to remove from the tissue. This is described below.

**Base multicellular model.** We use a modified version of the three dimensional multicellular model in Miller et al. [35]. This model uses an overlapping spheres methodology, where cells interact with each other via adhesion and repulsion forces, and with the basal membrane. A three dimensional model is used to avoid unnecessary artefacts of two dimensional hexagonal packing which can cause synchronisation of cell removal due to geometric constraints. Cell movement is determined from the inertia-less force balance:

$$\eta \frac{d\mathbf{c}_i}{dt} = \sum_{j \in N_i} \mathbf{F}_{ij} + \mathbf{F}_i^{\mathbf{Rt}} + \mathbf{F}_i^{\mathbf{D}}, \tag{30}$$

where $\mathbf{c}_i$ is the cell centre location of cell $i$; $N_i$ is the set of neighbours of cell $i$; $\mathbf{F}_{ij}$ are the forces on cell $i$ due to interacting neighbour cell $j$; $\mathbf{F}_{i,\text{ext}}$ are any external forces acting on cell $i$; and $\eta$ is the viscous drag coefficient for the cell [38]. A cell's neighbours are any cells within 2 cell diameters (distance between cell centres) of the cell of interest. $\mathbf{F}_i^{\mathbf{D}}$ is the desquamation force, shown in Fig 9 and described in a later section. $\mathbf{F}_i^{\mathbf{Rt}}$ is a rotational force, from Miller et al. [35], applied to the two daughter cells during division described below.

We define the cell-cell spring vector $\mathbf{r}_{ij} = r_{ij}\hat{\mathbf{r}}_{ij}$, for $r_{ij} = \|\mathbf{c}_j - \mathbf{c}_i\| - R_{ij}$ where $R_{ij}$ is the sum of the cell radii for cells $i$ and $j$, and $\hat{\mathbf{r}}_{ij}$ is the unit vector between the two cells. The intercellular adhesion and repulsion forces are based on functions from Atwell et al. [49] and Li et al. [34]

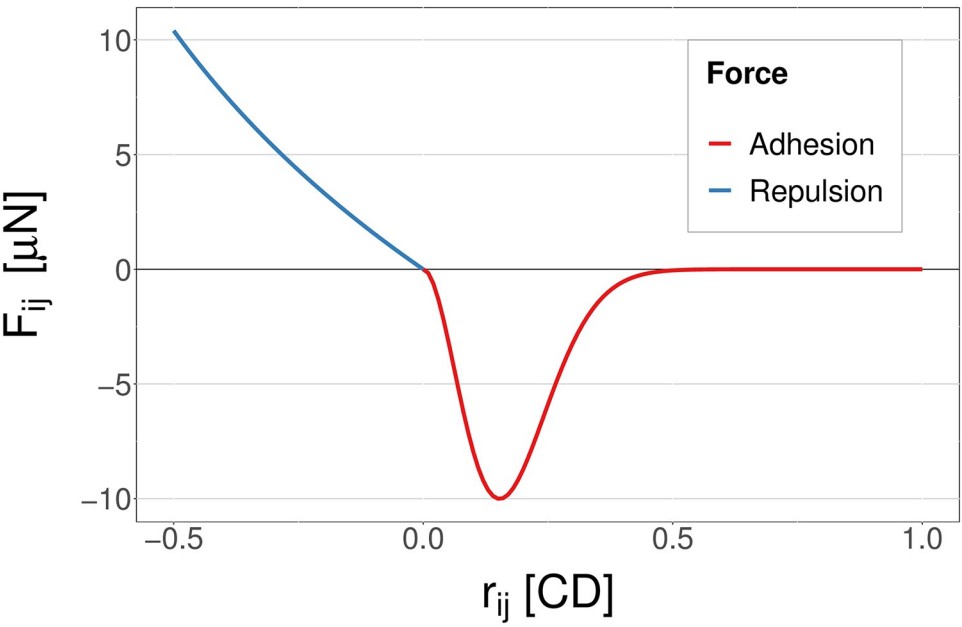

**Fig 8. The signed magnitude of the adhesive and repulsive forces ($F_{ij}$) experienced between two cells ($i$ and $j$) as a function of $r_{ij}$, the distance between the cell boundaries.** CD: Cell diameters.

respectively:

$$\mathbf{F}_{ij} = \begin{cases} -\alpha_{ij}\big((r^*_{ij} + c)e^{-\gamma(r^*_{ij}+c)^2} - ce^{-\gamma(r^{*2}_{ij}+c^2)}\big)\hat{\mathbf{r}}_{ij}, & r_{ij} > 0, \\ 0, & r_{ij} = 0, \\ k\,\log(1 + r_{ij})\hat{\mathbf{r}}_{ij}, & r_{ij} < 0, \end{cases} \tag{31}$$

$$\text{where } c = \sqrt{\frac{1}{2\gamma}}, \tag{32}$$

$$\text{and } r^*_{ij} = \frac{r_{ij}}{R_0}, \tag{33}$$

where $k$ is the repulsive spring constant [34, 49], $\alpha_{ij}$ is the adhesion coefficient between the two cells which is scaled by their CND (adhesion protein) level from the subcellular model (see Eq (39) below), and $R_0$ is the normal radii of the cells (0.5 cell diameters).

A plot of the adhesion and repulsion forces is shown in Fig 8. The adhesion force represents the effects of adhesion proteins between the cells: as the cells start to separate they experience forces from these proteins to pull them together, but as they separate further these adhesion proteins would begin to break apart. The repulsion force accounts for conservation of volume in the cells, and the steepness of its function is chosen to avoid instability in the model [45, 49].

The rotational force, $\mathbf{F}_i^{\mathbf{Rt}}$ in Eq (30), between two daughter cells, $i$ and $j$, during division is from Miller et al. [35] and is defined as:

$$\mathbf{F}_i^{\mathbf{Rt}} = -k_\phi \phi \hat{\mathbf{n}}, \tag{34}$$

where $k_\phi$ is a torsional spring constant (see Table 1); $\phi$ is the angle (in radians) between the

vertical direction, $\mathbf{k}$, and $\mathbf{r}_{ij}$; and $\hat{\mathbf{n}}$ is a unit normal to $\mathbf{r}_{ij}$ defined as:

$$\hat{\mathbf{n}} = \frac{\mathbf{r}_{ij} \times (\mathbf{k} \times \mathbf{r}_{ij})}{\| \mathbf{r}_{ij} \times (\mathbf{k} \times \mathbf{r}_{ij}) \|}. \tag{35}$$

This force is used to maintain the stem cell population in the basal layer. For further detail see Miller et al. [35].

The values of the parameters for the multicellular model are given in Table 1. Note, the normal adhesion coefficient for two cells below the corneum is much higher than that used in previous papers [34, 35] (374.7 $\mu$N rather than 0.2 $\mu$N), due to the removal forces applied to the system, as described below.

We are interested in homeostasis of the tissue, rather than tissue development, and so we need to run the model from an established tissue, as in Miller et al. [35]. To generate this established tissue we run a 'filling simulation': a simulation which begins with only the set of stem cells and populates a predefined tissue domain. For the duration of the filling simulation stem cells are restricted to the basal layer and cell removal occurs when cells move above a set (predefined) height. However, this predefined height is not the steady state thickness (i.e. thickness in homeostasis) of the system. Therefore, in order to ensure we are at steady state thickness, we also run the system for a burn-in period until it has reached a dynamic equilibrium (i.e. homeostasis), which is determined visually. Results are shown for 30 days of simulation, and therefore we set the minimum burn-in time to also be 30 days, giving a total simulated period of at least 60 days. We found, by visual inspection, that this is more than sufficient in all cases for the system to reach homeostasis. We only show the results for the final 30 days, and this is the period we use to calculate the steady state thickness.

Due to stochasticity in the proliferation cycle, 10 realisations are run for each simulation setup. This number is chosen as a balance between reliability of the results and the available computational resources (individual simulation times range from 42 to 58 hours depending on resulting tissue thickness). The horizontal boundaries are periodic, the base is bounded by the basal membrane, and the top boundary is defined by the desquamation process explained below. Further mechanisms are added to this model in order to model desquamation, and these are described in a later section.

**Rate parameter modifications for the multiscale model.** In order to produce a more computationally efficient system we decrease the number of cell layers in the tissue and use a higher proliferation rate. We note that increasing the proliferation rate is scaling the system in time, which enables us to reduce the required simulation time. An alternative to increasing proliferation rate might be to increase the time step, however due to the mechanical parameters of the system this is not feasible in this model as it leads to numerical instability.

The subcellular rate parameters ($k_{+1}$, $k_2$, $k_{+3}$, and $k_{-3}$) need to be adjusted to account for the size and time scalings. The fit for pH in Eq (22) is already normalised to the thickness of the corneum and therefore this does not need to be changed. The rate parameters are scaled as follows:

$$\hat{k}_i = \frac{\hat{T}_M}{T_M} k_i, \ \hat{T}_M = \frac{\tau_T}{v_z}, \tag{36}$$

where $\tau_T$ is the target corneum thickness, $v_z$ is the expected velocity of the cells given the proliferation rate, $\hat{T}_M$ and $T_M$ are the migration times of the multicellular and single cell systems respectively, and $\hat{k}_i$ and $k_i$ are the rate parameters for the multicellular and single cell systems respectively. We choose $\tau_T = 8$ CD which, given $T_M = 480$ hrs and $v_z \approx 0.05$ CD.hr$^{-1}$ from

Miller et al. [35], gives a migration time in the multicellular system of $\hat{T}_M = 160$ hrs. The new rate parameters for the enzyme-substrate reaction ($\hat{k}_{+1}$ and $\hat{k}_2$) are given in Table 1 and the new functions for the enzyme-inhibitor rate parameters are:

$$\hat{k}_{+3} = g_{+3}(\text{pH}) = (15.6\text{pH} - 58.5) \times 10^7 \ [\text{M}^{-1}.\text{hr}^{-1}], \tag{37}$$

$$\hat{k}_{-3} = g_{-3}(\text{pH}) = 6.9 \times 10^6 \ e^{-3.0\text{pH}} \ [\text{hr}^{-1}]. \tag{38}$$

**Combining the multicellular and subcellular model.**   The coupling and solution of the ODE model follows Algorithm 1. The subcellular degradation model is solved for each cell in the system for each multicellular time step using an adaptive ODE solver in the CVODE numerical library [50]. We set a relative tolerance of $10^{-4}$ and an absolute tolerance of $10^{-6}$ for the solver.

Coupling between the two scales occurs at the multicellular time step: every 30 seconds, as stated in Table 1. Both the cell location and current tissue height are taken from the multicellular model to determine pH, and consequently the rate parameters. We update the tissue height every 120 time steps, or once an hour (parameter 'Thickness calculation frequency' in Table 1), due to the high stochasticity in the height of the system, as seen in the Results section. A comparison to results in which the height is recalculated at every time step (results not shown) show negligible difference in the determined height. The output from the subcellular model into the multicellular system is the current proportion of $s$, or adhesion proteins, remaining. We ignore any anisotropy in adhesion degradation around the cell, as the driving forces we use to mimic the desquamation process are implemented in the vertical direction only. Cells in the first few layers of the tissue, $z < h$ for some specified $h$ (here $h = 4$), are considered to be in the lower layers of the tissue (not the corneum) and not experiencing adhesion degradation so for this region we set the ODE system to zero.

The multicellular model then needs to relate the proportion of adhesion proteins to the cell-cell adhesion function. The proportional loss of the proteins is assumed to be directly proportional to the loss of adhesion, as these proteins provide the cell-cell adhesion. In order to determine the adhesion between two cells we use the mean of the two protein concentrations:

$$\alpha_{ij} = \frac{1}{2}(s_i + s_j)\alpha_0, \tag{39}$$

where $s_i$ and $s_j$ are the remaining level of CND for the two cells, and $\alpha_0$ is the normal cell-cell adhesion coefficient.

**Algorithm 1**: Linking the subcellular model to cell-cell adhesion in the multicellular model

```
for each multicellular model time step do
  for every cell do
    Determine the parameters:

      1. Height in corneum given cell location (z) and current corneum
         thickness (τ), ξ = z/τ,
      2. pH = f_pH(ξ) using Eq (22),

      3. k̂_+3 = g_+3(pH) and k̂_-3 = g_-3(pH) using Eqs (37) and (38),

      4. k̂_+1 and k̂_2 from Table 1.
    Integrate Eqs (16) to (21) to next multicellular time step,
    Store s for the cell.
  end
  for every cell pair do
```

```
        Get s_i and s_j for the two interacting cells i and j,
        Calculate interaction force (Eq (31)), with adhesion coefficient
    scaled according to Eq (13).
      end
      for every cell do
        Calculate the rotational force for dividing cells,
        Add the desquamation force if surface cell,
        Determine new cell location using Eq (30).
      end
      Check if any cells have detached from the tissue and remove.
      Determine new set of surface cells that form the top layer of the
    tissue.
    end
```

## Modelling desquamation in the multiscale model

In order to model desquamation using the degradation of adhesion mechanism, we implement 'force-based cell removal': a 'removal force' is applied to the top cells in the tissue (surface cells), causing them to separate from the main tissue body. A similar force has previously been used by Li et al. [34]. Once a cell, or a set of cells, is completely separated from the main tissue body, it is removed from the simulation. This method can be broken down into three mechanisms: the definition of surface cells; the removal force; and the definition of separated cells. We address each of the three mechanisms individually below, and also show a two dimensional example of the method in Fig 9.

**Defining the surface cells.** First, we need to determine which cells are at the surface of the tissue. We do this by splitting the horizontal plane into a square grid with grid size $\Delta x$. We loop over the cell population and determine the highest cell (using the cell centres) in each grid square. This cell is labelled as a 'surface cell'.

Intuitively, one would calculate the surface density (cells per unit area) of the top layer to determine an appropriate value of $\Delta x$. However, the simulation output does not show clear

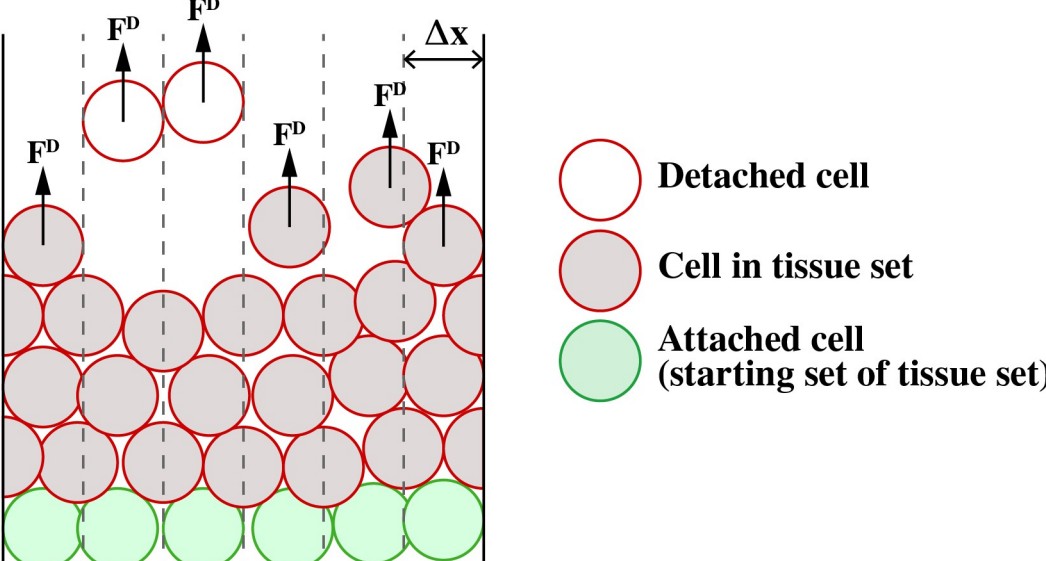

**Fig 9. The desquamation model.** Surface cells experience a vertical desquamation force. Once they detach from the tissue they get removed. Detachment is defined by determining the connected tissue set, starting with the set of attached cells.

layering, and therefore it is difficult to define a surface density. Close to the basal layer, where more layering is present, cell densities are around 1.2 cells/CD$^2$ (CD: cell diameters). However, after about the fourth layer, the cell arrangement is no longer obviously layered, becoming increasingly random, and cell density starts to decrease. Consequently, for simplicity and symmetry with proliferating cell count we use a grid size of $\Delta x$ = 1 CD. This means 100 cells are undergoing desquamation at any point in time, equal to the number of proliferating cells. Changing this grid density does not qualitatively change the results (results not included for brevity).

**Removal force.**    Once we have identified which cells are the surface cells we can then apply the removal force, $\mathbf{F}_i^D$ in Eq (30), as shown in Fig 9. The removal force is of set magnitude and is intended to reflect the environmental forces experienced by exposed skin cells. Though we would not expect the forces to be purely vertical in reality, we neglect any horizontal shear forces for the purposes of our model. What is important is that the bonds between cells are sufficiently broken to allow removal. The reduction in cell-cell adhesion is due to the degradation of the cell-cell bonds, which is known to occur earlier in the vertical direction than the horizontal direction [16]. Consequently, we believe a vertical force is a good approximation of this process.

**Detached cells.**    Once the force is applied to the surface cells, they start to separate from the tissue and we need to define when cells are no longer connected to the main tissue body. We do this by determining the cell set that constitutes the main tissue body. Starting with a set of cells which we know are in the main body, we can determine which cells are in contact with this set of cells. We iteratively add the contacting cells to the set, and repeat this process until no new cells are added to the set. This is then our main tissue body, and any cells not in the set are considered separated, and consequently removed. We use the set of cells attached to the basal membrane as our initial set.

In order for prompt removal of the cells, such that the removal process has minimal effect on the tissue thickness, we use a removal, or desquamation, force of $\mathbf{F}_i^D = 5\ \mu N$. In the presence of no other forces, a cell experiencing this force would be removed from the tissue in 1.4 hours (9% of the cell cycle). However, as a result of using this increased force for removal, an increased adhesion force is required to counter the effects of this force until the adhesion is sufficiently degraded. For simplicity, we set our adhesion force to twice that of the removal force: $\max(\mathbf{F}_{ij}(r_{ij} \geq 0)) = 10\ \mu N$. We note that this is not the same as setting $\alpha_0 = 10\ \mu N$ in the adhesive function in Eq (31)—the equivalent coefficient value is $\alpha_0 = 374.7\ \mu N$. Changing this relationship between the expected removal forces and cell-cell adhesion force coefficient would change the point at which the removal occurs during the degradation function, however this would not qualitatively change the results.

## Chaste implementation

The full multiscale model is implemented using the Chaste simulation libraries [46, 51, 52] for cardiac and multicellular tissue simulations. Chaste is a C++ library and the core code can be found at https://chaste.cs.ox.ac.uk/trac/wiki. To reproduce the model and results in this paper, further code and documentation, as well as all results data, can be found at https://github.com/clairemiller/MultiscaleModellingDesquamationInIFE.

## Supporting information

**S1 Text. A. Parameter determination for subcellular model**. Sourcing and fitting the parameters for the subcellular model. **B. Analysis of the enzyme system**. Extended results for the subcellular model. **C. Distinct proliferative cell niches can be represented by a**

**homogeneous population in the multiscale model**. Multiscale model results for two proliferative populations with different cell cycle lengths.
(PDF)

**S1 Video. Normal tissue video.** A video of one realisation of the healthy system (shown in Fig 4), with cells coloured by (left) the proportion of remaining corneodesmosome, $s$, and (right) the proportion of free KLK enzyme.
(MOV)

**S2 Video. Abnormal inhibitor levels tissue video.** A video of one realisation of the system (shown in Fig 5) with no inhibitor present, with cells coloured by (left) the proportion of remaining corneodesmosome, $s$, and (right) the proportion of free KLK enzyme.
(MOV)

**S3 Video. Treated tissue video $s$.** A video of one realisation of the treated system (shown in Fig 7) with half the cells recovered to normal inhibitor levels, and the other half with no inhibitor. Cells are coloured by (left) the proportion of remaining corneodesmosome, $s$, and (right) the proportion of free KLK enzyme.
(MOV)

## Acknowledgments

For author CM: the majority of the work for this study was undertaken during former positions at 1 School of Mathematics and Statistics, The University of Melbourne, Parkville, Australia, and 2 Systems Biology Laboratory, School of Mathematics and Statistics, and Department of Biomedical Engineering, The University of Melbourne, Parkville, Australia.

## Author Contributions

**Conceptualization:** Claire Miller, James M. Osborne.

**Formal analysis:** Claire Miller.

**Funding acquisition:** Edmund Crampin.

**Investigation:** Claire Miller, James M. Osborne.

**Methodology:** Claire Miller, Edmund Crampin, James M. Osborne.

**Supervision:** Edmund Crampin, James M. Osborne.

**Visualization:** Claire Miller.

**Writing – original draft:** Claire Miller.

**Writing – review & editing:** Claire Miller, James M. Osborne.

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
