## [Decision Letter · Decision Letter 0]

8 Dec 2021

Dear Dr. Osborne,

Thank you very much for submitting your manuscript "Multiscale modelling of desquamation in the interfollicular epidermis" for consideration at PLOS Computational Biology.

As with all papers reviewed by the journal, your manuscript was reviewed by members of the editorial board and by several independent reviewers. In light of the reviews (below this email), we would like to invite the resubmission of a significantly-revised version that takes into account the reviewers' comments.

Please particularly consider the reviewers questions on biological insights and importance of the proposed model. 

We cannot make any decision about publication until we have seen the revised manuscript and your response to the reviewers' comments. Your revised manuscript is also likely to be sent to reviewers for further evaluation.

Sincerely,

Pau Creixell

Guest Editor

PLOS Computational Biology

Douglas Lauffenburger

Deputy Editor

PLOS Computational Biology

Reviewer's Responses to Questions

**Comments to the Authors:**

Reviewer #1: Please see attached pdf

Reviewer #2: See attachment.

**Have the authors made all data and (if applicable) computational code underlying the findings in their manuscript fully available?**

Reviewer #1: Yes

Reviewer #2: Yes

PLOS authors have the option to publish the peer review history of their article (what does this mean?). If published, this will include your full peer review and any attached files.

Reviewer #1: No

Reviewer #2: No
---

## [Decision Letter · Decision Letter 1]

15 Apr 2022

Dear Dr. Osborne,

Thank you very much for submitting your manuscript "Multiscale modelling of desquamation in the interfollicular epidermis" for consideration at PLOS Computational Biology.

As with all papers reviewed by the journal, your manuscript was reviewed by members of the editorial board and by several independent reviewers. In light of the reviews (below this email), we would like to invite the resubmission of a significantly-revised version that takes into account the reviewers' comments.

We cannot make any decision about publication until we have seen the revised manuscript and your response to the reviewers' comments. Your revised manuscript is also likely to be sent to reviewers for further evaluation.

Sincerely,

Pau Creixell

Guest Editor

PLOS Computational Biology

Douglas Lauffenburger

Deputy Editor

PLOS Computational Biology

Reviewer's Responses to Questions

**Comments to the Authors:**

Reviewer #1: As far as I can tell, you have addressed the first half (roughly) of my comments and feedback, up to and including the point "Figure 8b – the difference between the two blue and red lines is unclear", but not the points that appear after this. I suspect that perhaps there has been an error in accessing the complete set of comments in my review?

**Have the authors made all data and (if applicable) computational code underlying the findings in their manuscript fully available?**

Reviewer #1: Yes

PLOS authors have the option to publish the peer review history of their article (what does this mean?). If published, this will include your full peer review and any attached files.

Reviewer #1: No
---

## [Decision Letter · Decision Letter 2]

8 Jul 2022

Dear Dr. Osborne,

We are pleased to inform you that your manuscript 'Multiscale modelling of desquamation in the interfollicular epidermis' has been provisionally accepted for publication in PLOS Computational Biology.

Best regards,

Pau Creixell

Guest Editor

PLOS Computational Biology

Douglas Lauffenburger

Deputy Editor

PLOS Computational Biology

Please consider incorporating these final couple of suggestions from reviewer 3.

Reviewer's Responses to Questions

**Comments to the Authors:**

Reviewer #1: Thank you for addressing all my original comments in this revision. I understand that it was a technical issue relating to transcription in the previous response.

Reviewer #3: Fig. 8 - magnitude of force is positive definite.

I find the discussion of other cell-based models in the literature rather long-winded now. It would be better to acknowledge these papers more succinctly in the intro and then critique in the discussion (para starting on L127 requires me to know about your model, which I have not yet been introduced to).

**Have the authors made all data and (if applicable) computational code underlying the findings in their manuscript fully available?**

Reviewer #1: Yes

Reviewer #3: Yes

PLOS authors have the option to publish the peer review history of their article (what does this mean?). If published, this will include your full peer review and any attached files.

Reviewer #1: No

Reviewer #3: No

---

## [Editor Report · Acceptance letter]

17 Aug 2022

PCOMPBIOL-D-21-01409R2 

Multiscale modelling of desquamation in the interfollicular epidermis

Dear Dr Osborne,

I am pleased to inform you that your manuscript has been formally accepted for publication in PLOS Computational Biology. Your manuscript is now with our production department and you will be notified of the publication date in due course.

With kind regards,

Anita Estes
